# T cell receptor clonotypes predict human leukocyte antigen allele carriage and antigen exposure history
Hesham ElAbd [1,2,4], Aya K. H. Mahdy [1,4], Eike Matthias Wacker [1], Maria Gretsova[1], David Ellinghaus [1], Astrid Dempfile [3] & Andre Franke [1] ✉

Conventional T cells recognize peptides presented by the human leukocyte antigen (HLA) proteins through their T cell receptors (TCRs). Given that thousands of HLA proteins have been discovered, each presenting thousands of different peptides, decoding the cognate HLA protein of a TCR experimentally is a challenging task. To address this problem, we combined statistical learning methods with a unique dataset of paired T cell repertoires and HLA allotypes for 6,794 individuals. This enabled us to discover 34,206 T cell receptor alpha (TRA) and 891,564 beta (TRB) clonotypes that were associated with 175 unique HLA alleles. The identified clonotypes target prevalent infections, *e.g.* influenza, cytomegalovirus and Epstein-Barr virus. Utilizing these clonotypes, we develop statistical models that impute the carriership of common HLA alleles from the TRA- or the TRB- repertoire. In conclusion, the identified allele-associated clonotypes encode the HLA fingerprints and the antigenic exposure history of individuals and populations.

T cells are an essential component in fighting infections and controlling cancers[1] as well as mediating tissue homeostasis[2]. T cells are characterized by a high diversity in their receptors, forming the basis of their ability to recognize a broad spectrum of antigens. Based on the nature of the antigens they recognize, T cells are divided into two main categories: conventional T cells which recognize peptides presented by the human leukocyte antigen (HLA) proteins and unconventional T cells which recognize a wide array of non-peptide antigens presented by non-HLA molecules such as vitamin B metabolites presented by the MR1 protein[3], and lipids presented by the CD1 molecules[4]. T cell receptors (TCRs) are heteromeric proteins that are made from two antigen-binding chains. In humans, there are four antigen binding chains, namely α, β, γ, and δ, where α and β preferentially dimerize together to form αβ TCRs, γ and δ chains bind together to form γδ TCRs. Conventional T cells utilize αβ TCRs while unconventional T cells utilize both αβ TCRs such as mucosal-associated invariant T (MAIT) cells, which recognize MR1-presented antigens[3], and γδ TCRs such as Vγ9Vδ2 T cells which recognize phosphoantigens presented by butyrophilins[5]. We here focused on conventional αβ TCRs.

Each α and β chain is generated through a somatic recombination process termed V(D)J recombination in which the recombination between different V genes, D genes (only in the β chains), and J genes coupled with the random insertion and deletion of nucleotides generates diverse antigen-binding chains. Recent advances in next-generation sequencing have advanced our ability to profile the collection of TCR antigen-binding chains present in a sample, *i.e,* the α (TRA) or the β (TRB) immune repertoire. Bulk TCR-Seq is a commonly used method to profile the immune repertoire by identifying the collection of the V(D)J recombination events constituting the repertoire[6]. Although TCR-Seq can be conducted using different assays and methods, it is commonly performed using a multiplex PCR that amplifies the DNA-encoded V(D)J recombination products, followed by deep sequencing and bioinformatics analyses to identify and quantify the expansion of different clonotypes in a given repertoire[6].

Given that different studies have shown that HLA alleles have a strong influence on the composition of TCR repertoires[7–9], subsequent studies aimed at associating T cell clonotypes with different HLA alleles. One of the earliest studies was performed by Emerson and colleagues, who used a statistical framework to analyze the TRB repertoire of 666 individuals with matched *HLA-A* and *HLA-B* allotypes, enabling them to identify thousands of clonotypes that were restricted to multiple *HLA-A* and *HLA-B* alleles[10]. This was extended in a follow-up work by DeWitt et al.[11] to include *HLA-C* alleles as well as class II proteins, *i.e., HLA-DR, HLA-DQ,* and *HLA-DP*. While these two studies were able to associate or link the carriership of a specific HLA protein with their corresponding public TRB clonotypes, they have not reported models that can impute HLA protein carriership from the

[1]Institute of Clinical Molecular Biology, Kiel University and University Hospital Schleswig-Holstein, Kiel, Germany. [2]Institute for Digestive Research, Lithuanian University of Health Sciences, Kaunas, Lithuania. [3]Institute of Medical Informatics and Statistics, Kiel University and University Hospital Schleswig-Holstein, Kiel, Germany. [4]These authors contributed equally: Hesham ElAbd, Aya K. H. Mahdy. ✉e-mail: a.franke@ikmb.uni-kiel.de

TRB or TRA repertoire. These models were later reported by Ortega et al.[12] who analyzed multiple publicly available datasets of paired HLA allotypes and TCR repertoires to develop models for imputing HLA alleles from the TCR repertoire. Lastly, Zahid et al.[13] described the analysis of paired TRB repertoires and HLA allotypes for ~4000 individuals and the development of a machine-learning framework for predicting the HLA background of an individual using its TRB repertoire.

These studies have established the feasibility of performing HLA typing from the T cell repertoire as well as the utility of large-scale statistical analyses in identifying public clonotypes associated with different HLA alleles. However, all mentioned studies, apart from Ortega et al.[12], did not release a publicly available tool for imputing HLA alleles from TCR repertoires. As T cell repertoire analyses became an important method to understand tumors[14,15], responses to infectious diseases[16,17], and identifying antigens driving chronic inflammatory diseases[18–20], the developed imputation models can decrease the cost of these analyses by removing the need to perform laborious and expensive wetlab-based HLA-typing. Furthermore, these models can impute only functional HLA proteins shaping the T cell repertoire, which is important for *HLA-DQ* and *HLA-DP* proteins because both the α and the β chains are polymorphic; however, not all assembled αβ heterodimers can produce a functional protein. Lastly, the generated datasets of TRA and TRB clonotypes associated with their cognate HLA alleles are of paramount importance for analyzing disease-associated clonotypes, which are a group of public clonotypes that are expanded in a specific disease[18], for example, primary sclerosing cholangitis (PSC)[19] or inflammatory bowel disease (IBD)[20–22]. To produce publicly available tools to impute HLA allotypes from either the TRA or the TRB repertoire and to generate a database of public TCRs with their cognate HLA allotypes, we assembled a large dataset of 1240 TRA and 5554 TRB repertoires with their HLA allotypes, enabling us to identify the cognate HLA alleles for 34,206 TRA and 891,564 TRB clonotypes and 175 common HLA alleles. After that, we utilized the identified TRA and TRB clonotypes to develop machine-learning models that can be used to impute the carriership of these HLA alleles.

## Results

### Assembly of a large dataset of paired T cell repertoires and HLA allotypes

Given that sample size significantly influences the performance of the imputation models, we assembled a large dataset of paired TCR repertoires and HLA genotypes (Table 1). The TCR repertoires were generated using two TCR-Seq methods, namely, the ImmunoSEQ assay (Adaptive Biotechnologies) and the αβ TCR profiling assay (MiLaboratories), while HLA

alleles were imputed from dense SNP genotyping arrays[23] (Methods). This enabled us to build, to the best of our knowledge, the largest dataset of paired T cell repertoires with HLA alleles at the time of writing. This dataset included 433 unique HLA alleles from three different populations, namely, Germany, Norway, and the USA. Subsequently, we split the dataset into three parts, first, paired TRB repertoires and HLA alleles ($n = 5554$ pairs; Fig. 1a), second, paired TRA repertoires and HLA alleles ($n = 385$ pairs; Fig. 1a). The TCR repertoire of these two subsets was generated using the ImmunoSEQ assay. The third subset contains the TRA repertoires that were profiled using the αβ TCR profiling assay from MiLaboratories with matching HLA alleles ($n = 855$ pairs; Fig. 1a).

### Discovering public TRB clonotypes associated with different HLA alleles and developing HLA imputation models based on the TRB repertoire

After splitting the TRB-HLA dataset ($n = 5554$ pairs) into 80% training and 20% validation we used a similar framework to Zahid et al.[13] to build TRB-based HLA imputation models (Methods; Fig. 1). Briefly, we started by discovering clonotypes associated with HLA alleles using the statistical framework described by Emerson et al.[10]. For each allele, the frequency of public clonotypes in allele carriers relative to non-carriers was compared using a one-sided Fisher's exact test where clonotypes with an association *P*-value $< 1 \times 10^{-4}$ are designated as clonotypes associated with the respective HLA allele (Fig. 1b). Subsequently, we used the L1-regularised linear regression model described by Zahid et al.[13] to resolve the cases where the same clonotype is associated with multiple HLA alleles due to linkage-disequilibrium (Fig. 1b). Then, we used a weighted sum of the expansion of allele-associated clonotypes as well as the repertoire depth to develop a linear regression model that can distinguish carriers of this allele from non-carriers[13] (Fig. 1c). Lastly, each of these allele-models is provided with either the TRA or the TRB repertoire of a given sample to calculate the probability of carrying a given HLA allele (Fig. 1d).

Using 80% of the data, *i.e.*, the training dataset, we identified 722,060 clonotypes that were associated with 312 HLA alleles, with the number of associated clonotypes being a function of allele frequency (Fig. 2a–f). Within the *HLA-A* locus, only eight alleles had a carriership frequency above 5%, with the most frequent HLA alleles being HLA-A*02:01 with a carriership frequency of ~50% followed by the HLA-A*01:01 (Fig. 2a). Given the higher diversity at the *HLA-B* locus, there were 13 alleles with a carriership frequency above 5%, with the HLA-B*08:01 and the HLA-B*07:02 being the most frequent and the ones with the highest number of associated clonotypes (Fig. 2b). A similar pattern was observed at the *HLA-C* locus where

**Table 1 | Overview of datasets used to build HLA imputation models from TRA or TRB repertoires**

| Dataset name | Locus | Number of samples | Country | Phenotype | TCR-Seq method |
|---|---|---|---|---|---|
| HC 1 | TRB | 773[19] | Germany | Healthy blood donors | Adaptive ImmunoSEQ |
| CD 1 | TRB | 1186[20] | Germany | CD patients | Adaptive ImmunoSEQ |
| UC 1 | TRB | 480[20] | Germany | UC patients | Adaptive ImmunoSEQ |
| PSC 1 | TRB | 431[19] | Germany | PSC patients | Adaptive ImmunoSEQ |
| CD 2 | TRB | 1809[20,22] | USA | CD patients | Adaptive ImmunoSEQ |
| UC 2 | TRB | 854[20,22] | USA | UC patients | Adaptive ImmunoSEQ |
| IBD-U1 | TRB | 21[20,22] | USA | IBD-U patients | Adaptive ImmunoSEQ |
| HC 2 | TRA | 165[21] | Germany | Healthy blood donors | Adaptive ImmunoSEQ |
| CD 3 | TRA | 124[21] | Germany | CD patients | Adaptive ImmunoSEQ |
| UC 3 | TRA | 96[21] | Germany | UC patients | Adaptive ImmunoSEQ |
| HC 3 | TRA | 264[21] | Norway | Symptomatic controls | MiLaboratories kits |
| CD 4 | TRA | 231[21] | Norway | CD patients | MiLaboratories kits |
| UC 4 | TRA | 360[21] | Norway | UC patients | MiLaboratories kits |

HC refers to healthy controls, CD to individuals with Crohn's disease, UC to individuals with ulcerative colitis, and IBD-U to individuals with unclassified inflammatory bowel disease. Lastly, symptomatic controls represent individuals with symptoms of inflammatory bowel disease, but their endoscopy and lab tests failed to unambiguously diagnose CD or UC.

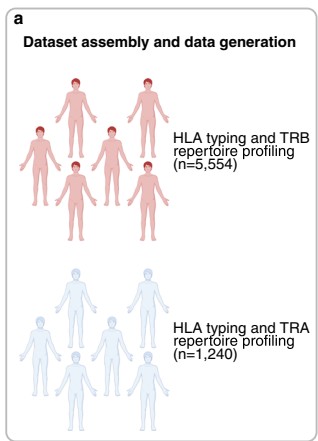
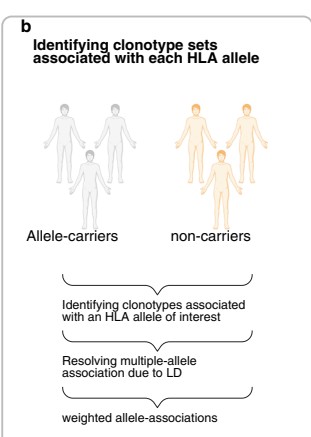
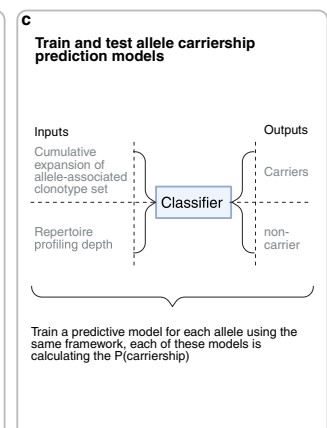
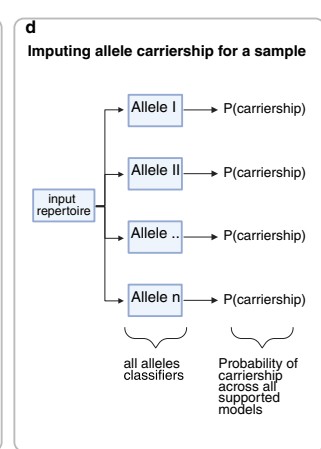

**Fig. 1 | Overview of the approach used for discovering clonotypes restricted to different HLA proteins and for developing models to impute the carriership of these HLA alleles based on the TRA or the TRB repertoire. a** Shows the cohorts used in the current study to discover TRA- and TRB- clonotypes associated with different HLA alleles. **b** Summarizes the discovery of clonotypes associated with each allele by comparing their presences in carriers and non-carriers using the Fisher's exact test followed by resolving linkage-disequilibrium (LD) using L1-regularised linear regression (L1LR)-models. **c** The machine-learning classifiers developed to predict the carriership of a given HLA allele using the cumulative weighted expansion of clonotypes that are associated with this HLA allele. **d** The pipeline for imputing HLA alleles from a given TRA or TRB repertoire, where for each of the supported allele-models we calculated the carriership probability. The final HLA-typing for a sample represents alleles with a carriership probability of 0.5 or more. *Created in BioRender. ElAbd, H. (2025)* https://BioRender.com/2gfo7ks.

only eleven alleles had a carriership frequency >5%, with the HLA-C*07:01, HLA-C*07:02 and HLA-C*06:02 being the most common and, consequently, the alleles with the highest number of associated clonotypes (Fig. 2c). Although the three HLA-I loci showed the same positive correlation between allele frequency and the number of associated clonotypes, the number of associated clonotypes was different among them, with the *HLA-B* locus having the highest number of associated clonotypes followed by the *HLA-A* locus and lastly the *HLA-C* locus (Fig. 2a–c).

Within the HLA-II alleles, the *HLA-DR* locus had the highest number of associated clonotypes, with only nine alleles having a carriership frequency above 5% with the HLA-DRB1*07:01 and HLA-DRB1*15:01 being the two *HLA-DR* alleles with the highest number of associated clonotypes (Fig. 2d). Within the *HLA-DQ* molecules, which are generated from the pairing of proteins encoded by the *HLA-DQA1* locus and the *HLA-DQB1* locus, nine *HLA-DQ* dimers had a carriership frequency > 5% (Fig. 2e). The most frequent *HLA-DQ* complex was derived from HLA-DQA1*01:02-DQB1*06:02, followed by the HLA-DQA1*01:01-DQB1*05:01 and the HLA-DQA1*05:01-DQB1*03:01 proteins (Fig. 2e). Only three *HLA-DP* complexes had a carriership frequency > 5% namely, HLA-DPA1*01:03-DPB1*02:01, followed by HLA-DPA1*01:03-DPB1*04:01 and lastly, HLA-DPA1*02:01-DPB1*04:01 (Fig. 2f).

Most of the clonotypes were restricted to HLA-II alleles ($n = 466,277$), particularly, HLA-DRB1 ($n = 303,330$) relative to all HLA-I alleles ($n = 145,224$), potentially, because of the higher ratio of CD4+ to CD8+ T cells in the blood. These findings also confirm previous reports by DeWitt et al.[11], specifically, (i) the strong positive correlation between allele frequency and the number of associated clonotypes, (ii) the higher number of clonotypes that are associated with HLA-II alleles, and (iii) the low number of clonotypes associated with the *HLA-C* locus. Using the L1-regularized logistic regression framework[13], we were able to resolve the association between clonotypes and multiple HLA alleles, however, for a small subset of clonotypes, this was not possible (Methods). Specifically, out of the 600,095 clonotypes that were associated with HLA alleles with a carriership frequency >5%, 587,224 (97.8%) clonotypes were associated with a single HLA allele while only 12,871 (2.2%) clonotypes were associated with multiple alleles.

After building prediction models for these 137 HLA alleles, we tested their performance on the 20% validation dataset ($n = 1111$ paired TRB-HLA repertoires). Starting with *HLA-A* alleles, we observed a high performance across most alleles with a median balanced accuracy of 0.88, median precision of 0.85 and a median recall of 0.77 (Supplementary Fig. 1). A similar trend was observed with *HLA-B* alleles where the median balanced

accuracy was 0.87, and the median precision and recall were 0.88 and 0.76, respectively (Supplementary Fig. 2). The performance of *HLA-C* allele models was lower than that of *HLA-B* and *HLA-A*, with a median balanced accuracy of 0.82, a median precision of 0.79, and a median recall of 0.66 (Supplementary Fig. 3). This might be attributed to the small footprint of *HLA-C* on the *TRB* repertoire as it has a lower surface expression[24] and a smaller immunopeptidome[25].

Regarding HLA-II alleles, *HLA-DR* alleles illustrated a high performance relative to *HLA-DQ* and *HLA-DP* alleles, with a median balanced accuracy of 0.89, a median precision of 0.89, and a median recall of 0.79 (Supplementary Fig. 4). *HLA-DQ* alleles had an average balanced accuracy of 0.60, an average precision of 0.46, and an average recall of 0.22 (Supplementary Fig. 5). While the average model performance was generally inferior to other HLA proteins discussed so far, some *HLA-DQ* models showed higher performance. For example, HLA-DQA1*01:02-DQB1*06:02 showed a balanced accuracy of 0.96, precision of 0.94 and a recall of 0.94 (Supplementary Fig. 5). A similar pattern was observed with *HLA-DP* alleles, where the average balanced accuracy was 0.67 and the average precision and recall were 0.58 and 0.35, respectively (Supplementary Fig. 6). These performance metrics were mainly driven by a handful of alleles that showed an accurate predictive performance such as the HLA-DPA1*01:03-DPB1*04:02 model, which had a balanced accuracy of 0.93, a precision of 0.92 and a recall of 0.89 (Supplementary Fig. 6).

To investigate factors influencing the performance of these models, we started by analyzing the impact of allele frequency and model performance. For *HLA-A* alleles we observed a positive correlation between the allele carriership frequency and different performance metrics (Fig. 2g). This relationship was not linear and showed signs of saturation, where carriership frequencies above 0.05-0.1 did not translate into a meaningful increase in the model performance. Similar findings were also observed for *HLA-B* (Fig. 2h), *HLA-C* (Fig. 2i) and *HLA-DR* (Fig. 2j) models. For *HLA-DQ* (Fig. 2k) and *HLA-DP* (Fig. 2l), this trend was also observed but to a lesser extent, with some alleles having relatively high carriership frequency ( > 0.1) but a relatively poor performance.

HLA-DQ and HLA-DP proteins are made from two different chains, α and β, leading to the formation of cis and trans proteins. Cis proteins are formed between chains encoded on the same chromosome, while the trans complexes are formed between chains encoded on different chromosomes. For example, an α chain encoded by the paternal copy and a β chain encoded by the maternal copy or vice versa. While the same applies to HLA-DR proteins, the α chain of the HLA-DR molecule is invariant, and

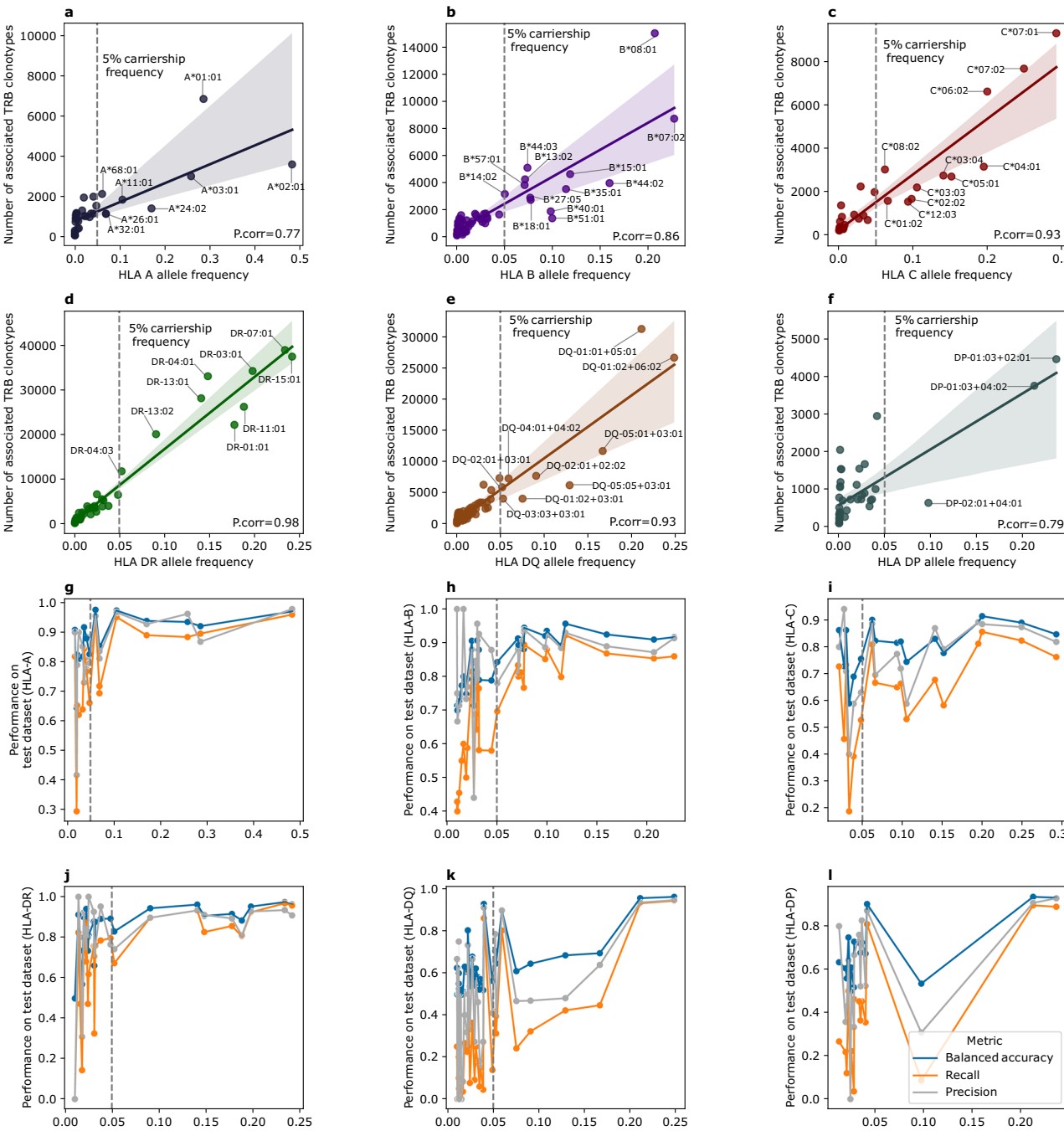

**Fig. 2 | Effect of HLA allele frequency on TRB-associated clonotypes and imputation model performance.** The relationship between HLA allele carriership frequency and the number of associated TRB clonotypes for the six classical HLA loci. *HLA-A* (**a**), *HLA-B* (**b**), *HLA-C* (**c**), *HLA-DR* (**d**), *HLA-DQ* (**e**), *HLA-DP* (**f**)."P.corr" denotes the Pearson correlation coefficient. For panel (**d**), *HLA-DR* alleles are written in the name of their corresponding *HLA-DRB1* alleles because the alpha chain is invariant, hence, DR-07:01 represents the HLA-DR molecules whose beta-chain is encoded by the HLA-DRB1*07:01 allele. For panels (**e**, **f**) HLA allele names are written as the alpha chain allele + the beta chain allele, for example, DQ-

01:01 + 05:01 represents the HLA molecules encoded by the HLA-DQA1*01:01 and the HLA-DQB1*05:01 alleles. Panels (**g–l**), the relationship between HLA-allele carriership frequency and the performance of its TRB-based imputation model on a test dataset of 1111 TRB repertoires with linked HLA allotypes. Three performance metrics were used to evaluate the model performance, namely, balanced accuracy, recall and precision. **g–i** The performance of three HLA-I loci models, namely, *HLA-A*, *HLA-B*, and *HLA-C*, respectively. Similarly, the performance of HLA-II molecules is illustrated in (**j–l**), with *HLA-DR* shown in (**j**), *HLA-DQ* in (**k**) and lastly, *HLA-DP* in (**l**). The data supporting panels (**g–l**) are provided in Supplementary data 1.

hence we focused only on the β chain located either on the paternal or on the maternal chromosome. Several studies have indicated that trans complexes have a minor impact on the formed immunopeptidome[26], as not all trans complexes lead to a stable molecule at the cell surface[27]. This might explain the poor performance of different *HLA-DQ* and *HLA-DP* complexes that have a relatively high carriership frequency, as these αβ

allele combinations might not generate a stable HLA complex and hence have a minor impact on the TRB repertoire.

To investigate this, we inferred *HLA-DQ* and *HLA-DP* haplotype structures statistically (Supplementary Fig. 7; Supplementary data 2 and Supplementary data 3) and compared the performance of cis and trans HLA-DP and HLA-DQ complexes (Supplementary Fig. 8). Most of the

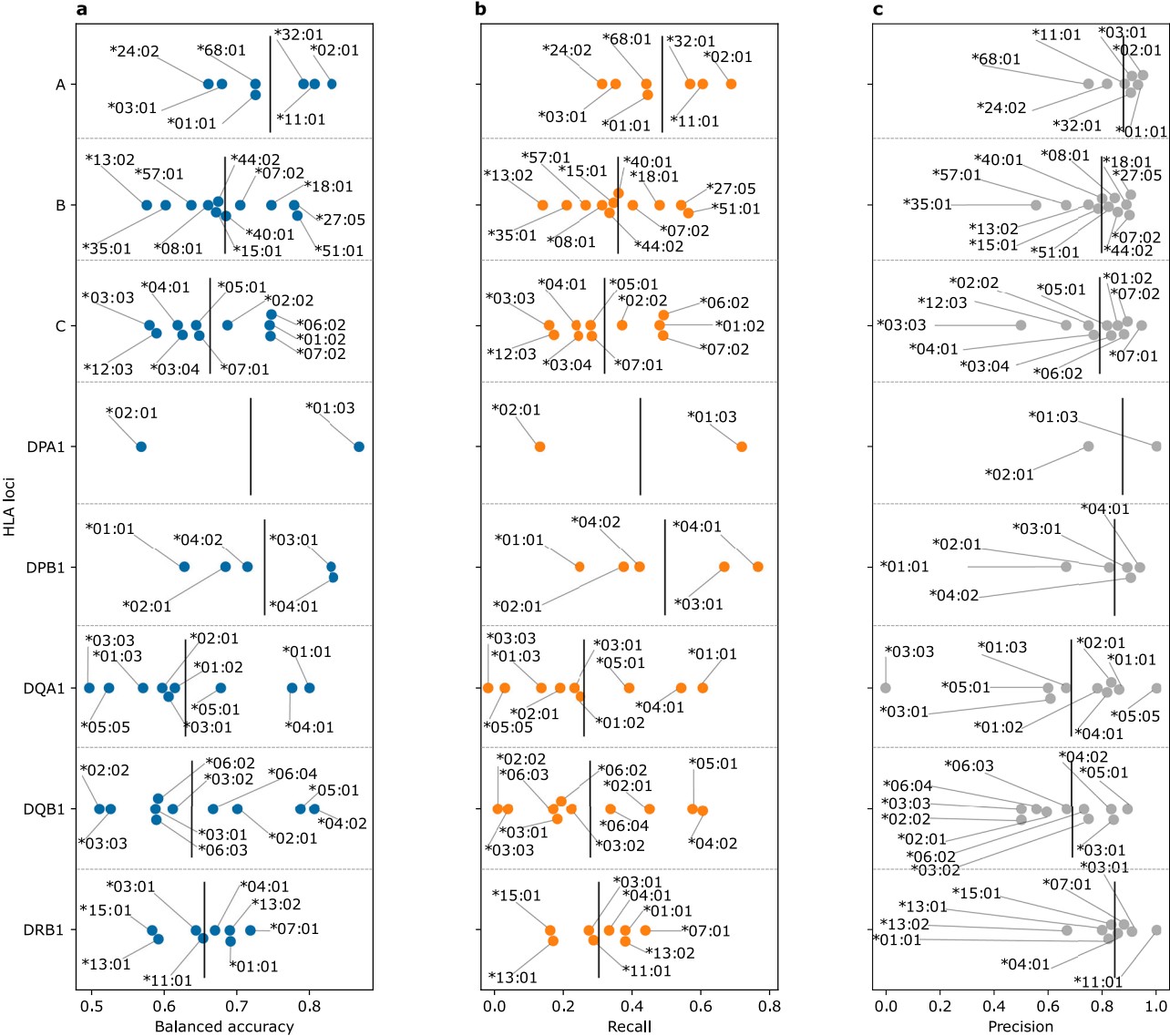

**Fig. 3 | The performance of the TRB-based HLA imputation models on an independent test dataset obtained from Rosati et al.[28]. a** shows the balanced accuracy, while (**b**) the recall and (**c**) the precision across different HLA alleles belonging to different HLA loci. Across all panels, alleles with carriership frequency <5% (n < 12 samples) were excluded from the analysis. The data supporting panels (**a–c**) are provided in Supplementary data 4.

alleles with a carriership frequency >1% were potentially cis complexes and we had models for only three trans HLA-DQ complexes, namely, HLA-DQA1*03:01-DQB1*03:03, HLA-DQA1*02:01-DQB1*04:02, and HLA-DQA1*05:05-DQB1*02:02. Similarly, for HLA-DP complexes only one trans-complex had a frequency >1%, specifically, HLA-DPA1*01:03-DPB1*16:01, but all other HLA-DP complexes (n = 15) were potentially cis. Thus, our observations suggest that not all potential cis *HLA-DP* or *HLA-DQ* complexes can be accurately imputed from the TRB repertoires.

Motivated by these findings and the performance of the models on the validation dataset, we used the entire dataset for training and developing imputation models using the same workflow introduced above. This enabled us to identify 1,095,576 unique TRB clonotypes that were associated with 437 unique HLA alleles, with 1,049,766 clonotypes showing single-allele association and 45,810 being associated with multiple alleles. After filtering for alleles with a carriership frequency >0.01 (corresponds to n > 55 individuals), we obtained 891,564 clonotypes that were associated with 175 HLA alleles, specifically: 17 *HLA-A* alleles, 27 *HLA-B*, 17 *HLA-C*, 22 *HLA-DR*, 30 *HLA-DP*, and 62 *HLA-DQ* alleles. Consistent with our previous observations, the number of associated clonotypes was higher for HLA-II alleles, particularly *HLA-DRB1* alleles, relative to HLA-I alleles. Despite the

strong differences between the different HLA loci, within a single locus, allele carriership frequency strongly correlated with the number of associated TRB clonotypes (Supplementary Fig. 9).

To test the performance of the developed models, we used a previously published dataset[28] of 229 healthy and IBD patients with paired HLA allotypes and immune repertoires, where we imputed the HLA of each sample using the TRB repertoire and compared the imputed results to the provided HLA allotypes. Furthermore, we focused the analysis on models with an allele carriership >5% in the training dataset. Although this test dataset was generated with a different TCR-Seq method and from RNA instead of DNA, our developed TRB-models were able to impute common HLA alleles (Fig. 3, Supplementary Fig. 10). Across the different loci, we observed a significant reduction in the recall (Fig. 3b, Supplementary Fig. 10), which might be attributed to differences in the TCR-Seq methodology used, as the models were trained on datasets generated via the ImmunoSEQ assay and tested on repertoires profiled using the MiLaboratories assay. The former assay generally enables a deeper repertoire profiling relative to the latter. Hence, this reduction in the profiling depth might explain the reduction in the ability of the models to recall alleles. Nonetheless, the precision of the models remained relatively unchanged (Fig. 3c and Supplementary Fig. 10).

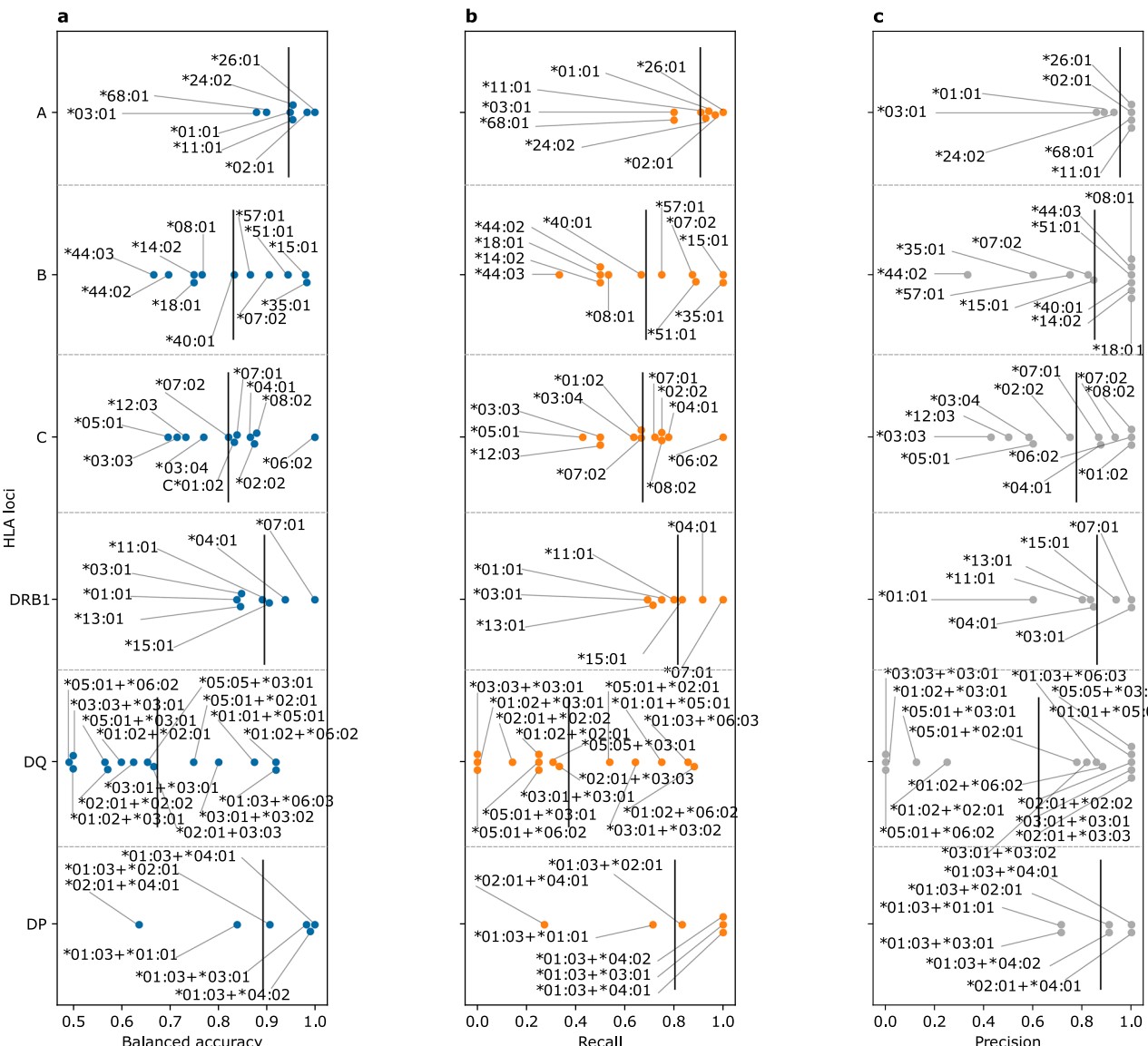

**Fig. 4 | The performance of the TRB-based HLA imputation models on an independent test dataset obtained from the immuneCODE dataset[29]. a** shows the balanced accuracy, while (**b**) the recall and (**c**) the precision across different HLA alleles belonging to different HLA loci. Across all panels, alleles with carriership frequency <5% ($n < 3$ samples) were excluded from the analysis. The data supporting panels (**a–c**) are provided in Supplementary data 5.

To test the performance of the models on an independent test dataset that was generated with the same technology, *i.e.*, the ImmunoSEQ assay, we used a subset of the immuneCODE[29] database with matching HLA allotypes ($n = 63$ individuals). Subsequently, we imputed the HLA allotypes of each sample from its TRB repertoire using the developed models and compared the imputed alleles to the reported HLA alleles. As seen in Fig. 4, most of the models showed an accurate performance, except for some HLA-DQ complexes (Supplementary Fig. 11). We observed a robust increase in the recall relative to the Rosati et al.[28] test dataset. Indicating that the decrease in the recall observed previously can be attributed to the shallow profiling, performed in the Rosati et al.[28] study.

**Discovering public TRA clonotypes associated with different HLA allotypes and developing HLA imputation models based on the TRA repertoire**

As the *TRA* repertoire was profiled using two different TCR-Seq technologies, namely, the ImmunoSEQ assay and the αβ TCR profiling assay from MiLaboratories, and using different starting materials, *i.e.*, DNA and RNA, respectively, we did not combine the two datasets and treated each dataset

independently. Starting with the dataset made from cohorts HC2, CD3, and UC3 which were profiled using the ImmunoSEQ assay from DNA (Methods), we split the dataset into an 80% training and a 20% validation datasets. We then used the framework described above to discover TRA clonotypes associated with HLA allotypes (Methods). Although our results mirrored those from the TRB repertoires, in which the number of associated clonotypes per HLA allele was strongly dependent on the allele's carriership frequency (Supplementary Fig. 12), the overall number of TRA-associated clonotypes was lower than the number of TRB-associated clonotypes. This can be attributed to differences in size between the two datasets ( > 5500 TRB repertoires vs. 308 TRA repertoires used for training, *i.e.*, ~ 6% of the TRB dataset), which highlights the impact of the dataset size on discovering clonotypes associated with each allele. Given the small number of repertoires and discovered allele associated-clonotypes, the resulting TRA-based imputation models showed a poor predictive performance, relative to the TRB-models, even for the common HLA alleles, *i.e.*, alleles with carriership frequency >5% (Supplementary Figs. 13– 18).

We repeated the same process with the other TRA-HLA datasets composed of HC3, CD4 and UC4 cohorts, which were profiled using the αβ

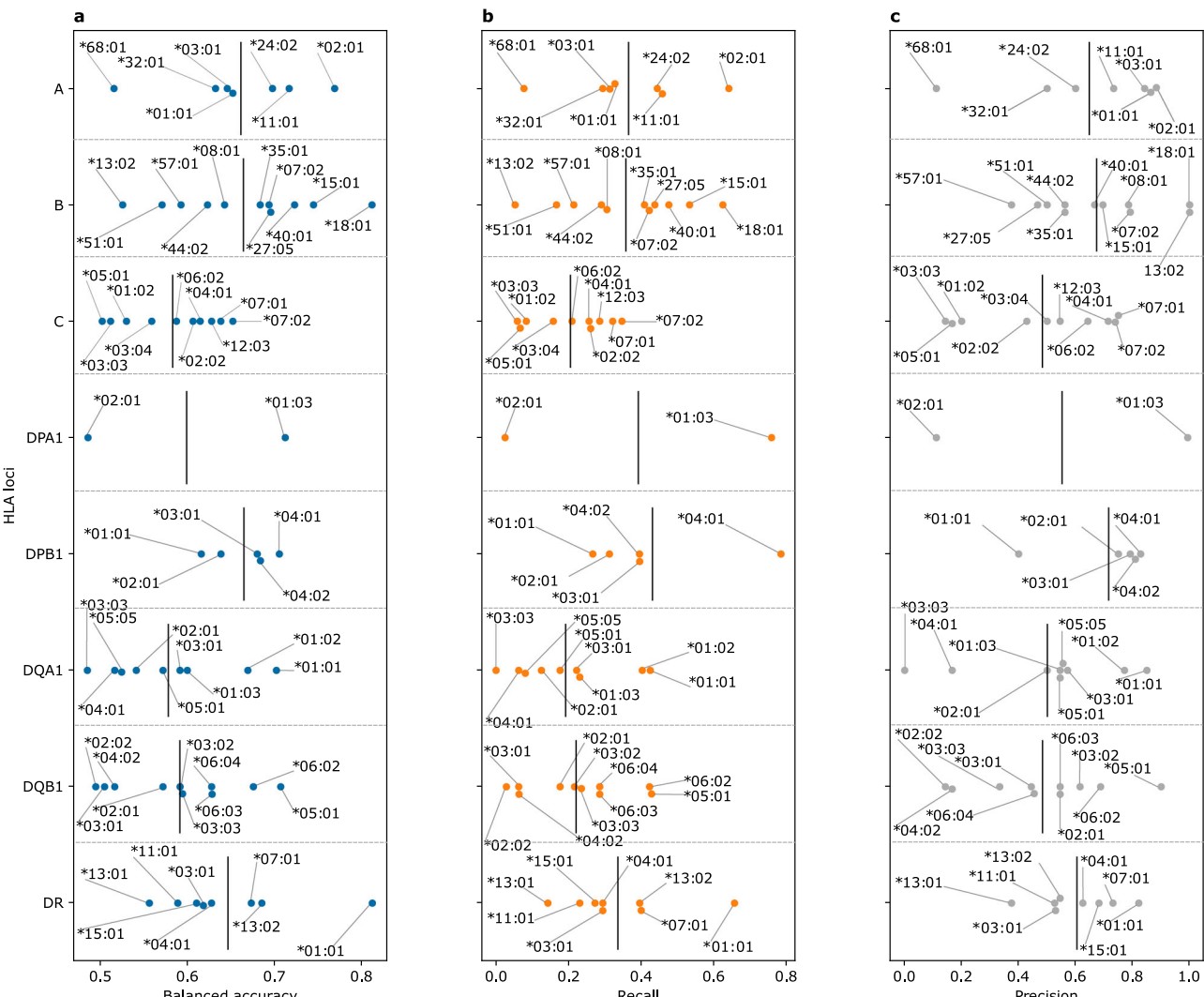

**Fig. 5 | The performance of the developed TRA-based imputation models on a test dataset of paired TRA repertoires and HLA allotypes that was generated by Rosati et al.[28]. a** shows the balanced accuracy, while (**b**) the recall and (**c**) the precision across different HLA alleles belonging to different HLA loci. Across all panels, alleles with carriership frequency <5% (*n* < 12 samples) were excluded from the analysis. The data supporting panels (**a–c**) are provided in Supplementary data 6.

TCR profiling assay from MiLaboratories with RNA as the starting biological material (Methods). Given that this dataset is ~two-fold the size of the previous TRA-HLA dataset, we observed a higher number of HLA-associated clonotypes, 31,230 relative to 9435 clonotypes. Similar to previous findings, the number of clonotypes associated with each HLA allele positively correlated with the carriership frequency (Supplementary Fig. 19). After testing the models on the 20% validation dataset, we observed a similar trend to the TRB findings where *HLA-A* (Supplementary Fig. 20) and *HLA-B* (Supplementary Fig. 21), showed, on average, a higher performance relative to *HLA-C* (Supplementary Fig. 22) across common HLA alleles (carriership frequency >5%). For common HLA-II alleles, *HLA-DR* alleles showed the highest performance (Supplementary Fig. 23) relative to *HLA-DQ* (Supplementary Fig. 24) and *HLA-DP* (Supplementary Fig. 25). Although the performance of these models was relatively higher than the first TRA-based models trained on the smaller TRA dataset obtained using the ImmunoSEQ assay, it is still inferior to the TRB-based models in terms of the number of supported HLA-alleles and the accuracy of each model.

Consequently, we focused the TRA-based model development on the larger dataset assembled from the HC3, the CD4 and the UC4 cohorts (n = 855 TRA-HLA pairs) where we used the entire dataset to develop imputation models for HLA alleles with a carriership frequency >5%. To test the generalizability of these models we used the independent Rosati et al.[28]

test dataset (Fig. 5, Supplementary Fig. 26). Relative to TRB-based imputation models, the TRA-models showed an inferior predictive performance, potentially due to the smaller training dataset, 855 TRA-HLA pairs, relative to 5,554 TRB-HLA pairs. Similar to TRB-based models, the TRA-based models suffered from a reduction in the recall relative to precision, potentially due to the shallow repertoire depth of the Rosati et al.[28] dataset.

## TCR2HLA achieves a state-of-the-art performance in imputing HLA allotypes from TRB repertoires

To evaluate the performance of the developed models against other TCR-based imputation pipelines, we benchmarked their predictive performance against HLAGuessr which was recently developed by Ortega et al.[12]. We focused on analyzing the performance of the TRB models. We selected the immuneCODE datasets[29] as it was not included in the training of our tool (TCR2HLA; Code availability) or that of Ortega and colleagues[12]. Besides the differences in predictive performance (Fig. 6), TCR2HLA offers two advantages over HLAGuessr, first, it can impute functional *HLA-DQ* and *HLA-DP* alleles and not just a chain-level prediction, *i.e.*, a specific *HLA-DQA* or *HLA-DQB* allele as HLAGuessr does. Second, it supports a larger number of alleles, specifically, 433 alleles, including >175 common HLA alleles, relative to 98 alleles supported by HLAGuessr. As HLAGuessr does not predict functional HLA-DQ and HLA-DP alleles, we restricted the

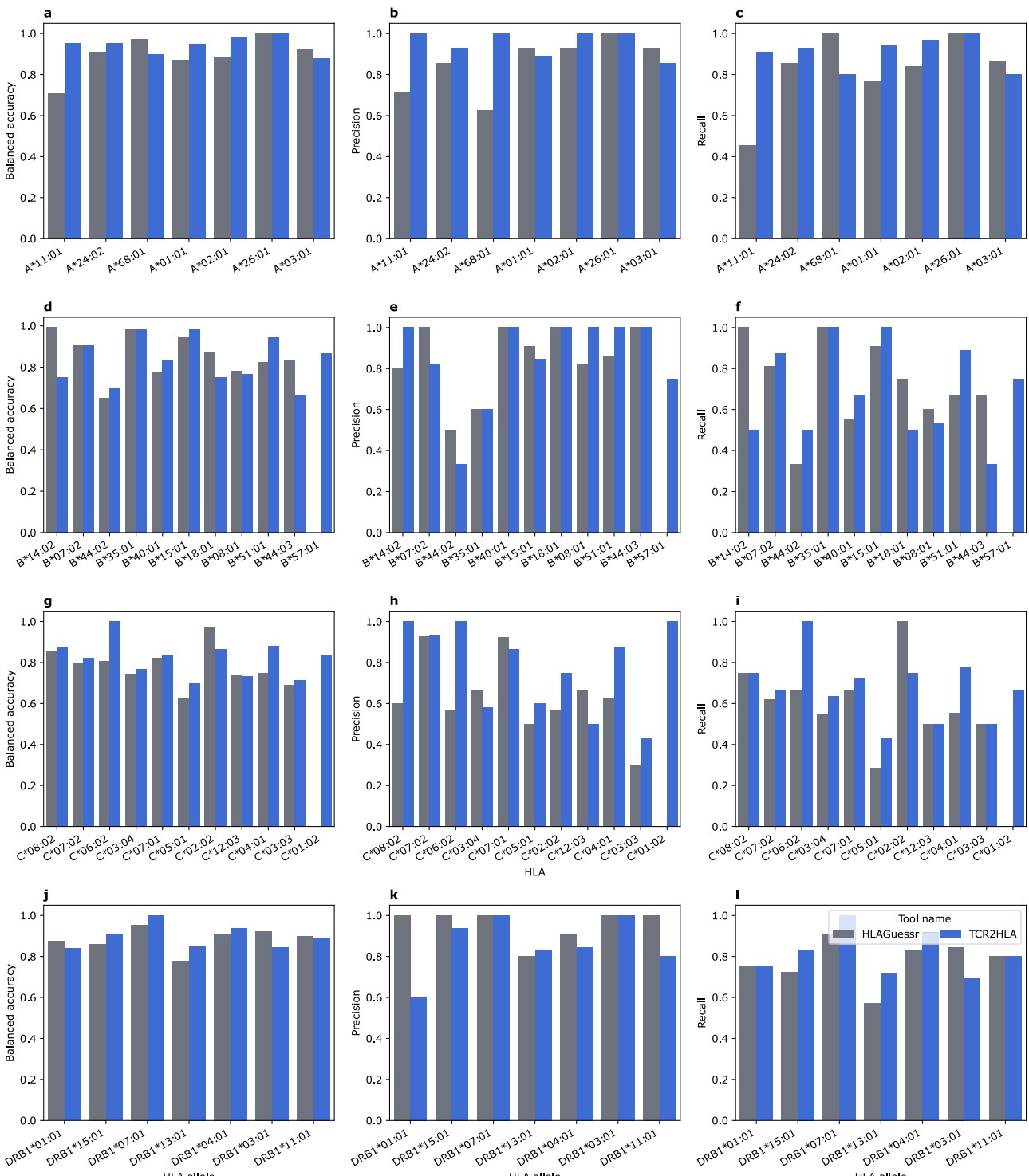

**Fig. 6 | Benchmarking the predictive performance of TCR2HLA against HLAGuessr[12] using the immuneCODE[29] dataset. a–c** The performance of *HLA-A* models across three metrics, namely, balanced accuracy, precision and recall, respectively. **d–f** The performance across the three evaluation-metrics for *HLA-B* models, (**g–i**) and (**j–l**) the benchmarking results for *HLA-C* and *HLA-DRB1* models, respectively. The supporting data is available in Supplementary data 7.

comparisons to the three HLA-I loci and the *HLA-DRB1* locus, focusing on common alleles. Across all loci, we observed a comparable performance, where for some alleles the performance of TCR2HLA was better while for others, HLAGuessr yielded a better performance, *e.g.*, HLA-A*11:01 and HLA-A*03:01, respectively (Fig. 6). Furthermore, the tools also differ across metrics, for example, the TCR2HLA model for the HLA-B*14:01 allele showed a higher precision while the HLAGuessr's model of the same allele showed higher recall and balanced accuracy (Fig. 6). Thus, for common

alleles, both tools achieved comparable performance. Consequently, future tools can be developed to combine predictions by both tools to improve the overall imputation accuracy of HLA alleles from TCR repertoire datasets.

## The discovered HLA-associated clonotypes target prevalent infections
To gain more insights into the identified HLA-associated TRA- and TRB-clonotypes, we analyzed their overlap with public TCR-antigen databases,

namely, McPAS[30] and VDJdb[31]. From public HLA-restricted TRA clonotypes identified from both TRA-HLA datasets, 1049 clonotypes overlapped with the assembled public database. Most of the identified clonotypes were restricted to common viral and bacterial infections such as cytomegalovirus (CMV), Epstein-Barr virus (EBV), influenza, and *M. tuberculosis*, and were specific toward HLA-I proteins with only 16 clonotypes (1.5%) restricted to HLA-II alleles. Most of these 1049 clonotypes were restricted to common *HLA-A* alleles, such as HLA-A*02:01 and HLA-A*03:01 (Fig. 7a). We could infer the antigenic specificity of only 1910 public HLA-restricted TRB clonotypes that were targeting prevalent infections such as CMV, EBV, and influenza. Furthermore, most of the identified clonotypes were restricted to common HLA-I alleles such as HLA-A*02:01, HLA-A*03:01, HLA-B*08:01 and HLA-B*07:02 (Fig. 7b). By analyzing the sequence of these TRA and TRB clonotypes (Methods; Fig. 7), we did not observe a significant degree of sequence similarity among them, suggesting that the identified clonotypes are recognizing different antigens presented by the same HLA protein.

## Discussion

We assembled a large dataset of TCR repertoires with their corresponding HLA allotypes which enabled us to discover thousands of clonotypes that are associated with hundreds of HLA alleles. Based on these clonotypes we were able to develop robust statistical models for imputing common HLA alleles. While previous studies focused mainly on analyzing the relationship between the TRB repertoire and HLA proteins[10,13], using 1240 TRA-HLA pairs, we illustrated that the TRA repertoire can also be used to impute HLA alleles. Although we assembled a large TRA-HLA dataset, it is still relatively small (<25%) compared to the TRB-HLA dataset. This also explains the inferior performance of TRA-based relative to TRB-based imputation models, highlighting the critical impact of dataset size on the predictive performance of these imputation models and generally on studying the interplay between TCRs and HLA proteins.

Besides the size of the dataset, the utilized TCR-Seq method had an impact on the imputation results. TCR-Seq can be conducted using different methods such as 5'-RACE, and multiplex PCR, and using different quantities of DNA or RNA[6], all of which have an impact on shaping the identified repertoires[32]. The performance of the developed imputation models depends on the utilized TCR-Seq dataset as the models showed superior performance with the immuneCODE test dataset[29] relative to the dataset reported by Rosati et al.[28]. Both of these datasets were profiled with different technologies that have different sensitivities, and profile the repertoire at a different depth. Lastly, both of our TRA- and TRB-based imputation models are based on the blood repertoire. We have recently shown systematic differences in the composition of the blood repertoire relative to tissues[33]. Thus, the performance of these models on the immune repertoire of tissues would vary based on the similarity of the tissue repertoire to that of the blood repertoire.

Despite the large size of the TRB-HLA dataset utilized in the current study, we were able to develop accurate models for HLA proteins that are common among European and Northern American populations only. This is because the performance of each of these models is strongly dependent on the corresponding HLA allele frequency in the training dataset. The relationship between allele frequency and different performance metrics followed a dose-response curve with a marginal improvement in performance once the allele carriership frequency was >10%. Thus, building models, or even discovering associated clonotypes, for rare HLA alleles was not possible. These rare HLA alleles might exhibit a higher frequency in different populations, for example, HLA-DRB1*15:02 and HLA-DRB1*15:03 alleles are rarer in European populations but higher in Asian and African American populations, respectively[34]. Selective sampling can be used to mitigate this problem, where the TCR repertoire is preferentially profiled for individuals with rare HLA-alleles, to provide a cost-effective way to extend the number of supported HLA alleles.

Another potential limitation to our study is that most of the individuals included were suffering from different forms of inflammatory diseases, mainly IBD. Our dataset is therefore enriched for known disease-associated HLA alleles such as HLA-DRB1*15:01[34] and HLA-DRB1*07:01[35], however, these alleles are also common in healthy individuals. A second limitation is using genotyping-based HLA-imputation to generate a paired TCR-HLA database for training and for developing TRA- and TRB- based HLA imputation models. Given that we focused only on common alleles, for which the genotyping-based imputation is fairly accurate[23], the noise introduced in the TCR-HLA association database should be minimal.

While HLA allele frequency was an important factor in determining the number of clonotypes associated with each HLA allele, there were notable differences among the HLA proteins encoded by different loci. *HLA-A*, *HLA-B* and *HLA-DR* alleles had on average a higher number of associated clonotypes relative to *HLA-C*, *HLA-DQ* and *HLA-DP* alleles. Different factors may contribute to this such as the level of surface expression where *HLA-C* has been shown to have a lower surface expression relative to other HLA-I proteins[24]. Similarly, *HLA-DP* and *HLA-DQ* have a lower surface expression relative to *HLA-DR* proteins[36]. Beyond the surface expression, the complexity and diversity of the immunopeptidome of these different HLA alleles have been shown to vary considerably, for example, the complexity of the *HLA-C* immunopeptidome is lower than that of *HLA-A* and *HLA-B* immunopeptidomes[25]. These findings indicate, at least from the T cell perspective, that antigen presentation by *HLA-A*, *HLA-B* and *HLA-DR* plays an important role in shaping adaptive immune responses relative to other HLA proteins.

Although we could not resolve the antigenic specificity for all HLA-associated clonotypes using public databases[30,31], we were able to annotate the antigenic specificity for >1000 TRA and TRB clonotypes. Most of these clonotypes were specific to common viral and bacterial infections. Similar observations were detected by analyzing the antibody repertoire of large cohorts using high throughput assays such as phage-immunoprecipitation sequencing (PhIP-Seq)[37–39]. In these studies, most public responses were against common viral and bacterial antigens such as EBV, CMV and other common bacterial infections[37–39]. Hence, these findings indicate that common bacterial and viral infections are a major determinant of the immune repertoire at both the cell-mediated and humoral immune responses. While the HLA influences humoral immune responses[40], its effect is much stronger on the cell-mediated immune responses because of HLA-restriction. Thus, the identified allele-associated clonotypes represent a fraction of the immune memory toward prevalent infections.

Mechanistically, the interaction between infections and HLA proteins will lead to the formation of different HLA-peptide complexes in different individuals, these complexes will be recognized by different T cells. Hence, carriers of a given allele will have a shared response in terms of T cell clonotypes toward the same HLA-peptide complex relative to individuals with different HLA alleles. Thus, the subset of allele-associated clonotypes present in an individual represent a fraction of the immune exposure history of this individual. By decoding the antigenic specificity of these clonotypes, a better understanding of the immunological history of an individual and/or a population can be developed. Alterations in responses to some of these common infections have been associated with different diseases, for example, inappropriate immune responses toward Epstein-Barr virus have been associated with multiple autoimmune and chronic inflammatory diseases such as multiple sclerosis[41], rheumatoid arthritis[42], Sjögren syndrome[43], and primary sclerosing cholangitis[19]. Additionally, cytomegalovirus has been linked with the adult-onset Still's disease[44], and human papillomaviruses with cervical cancers[45]. By analysing the immune repertoire of affected individuals and appropriate controls, we could identify a group of clonotypes that are expanded in affected individuals. These clonotypes will be restricted to specific HLA alleles and recognize a prevalent antigenic exposure within the affected individuals[18–20]. Hence, by decoding the antigenic exposure of these disease-associated and HLA-restricted clonotypes we can decode the antigenic exposure trajectory involved in a particular disease.

**Fig. 7 | The antigenic specificity of HLA-associated TRA- and TRB-clonotypes.** (**a**) depicts the overlap between HLA-associated TRA clonotypes and public databases, namely, VDJdb[31] and McPAS[30] while (**b**) illustrates the overlap between these databases and HLA-associated TRB clonotypes. Network visualization was performed using Cytoscape[57].

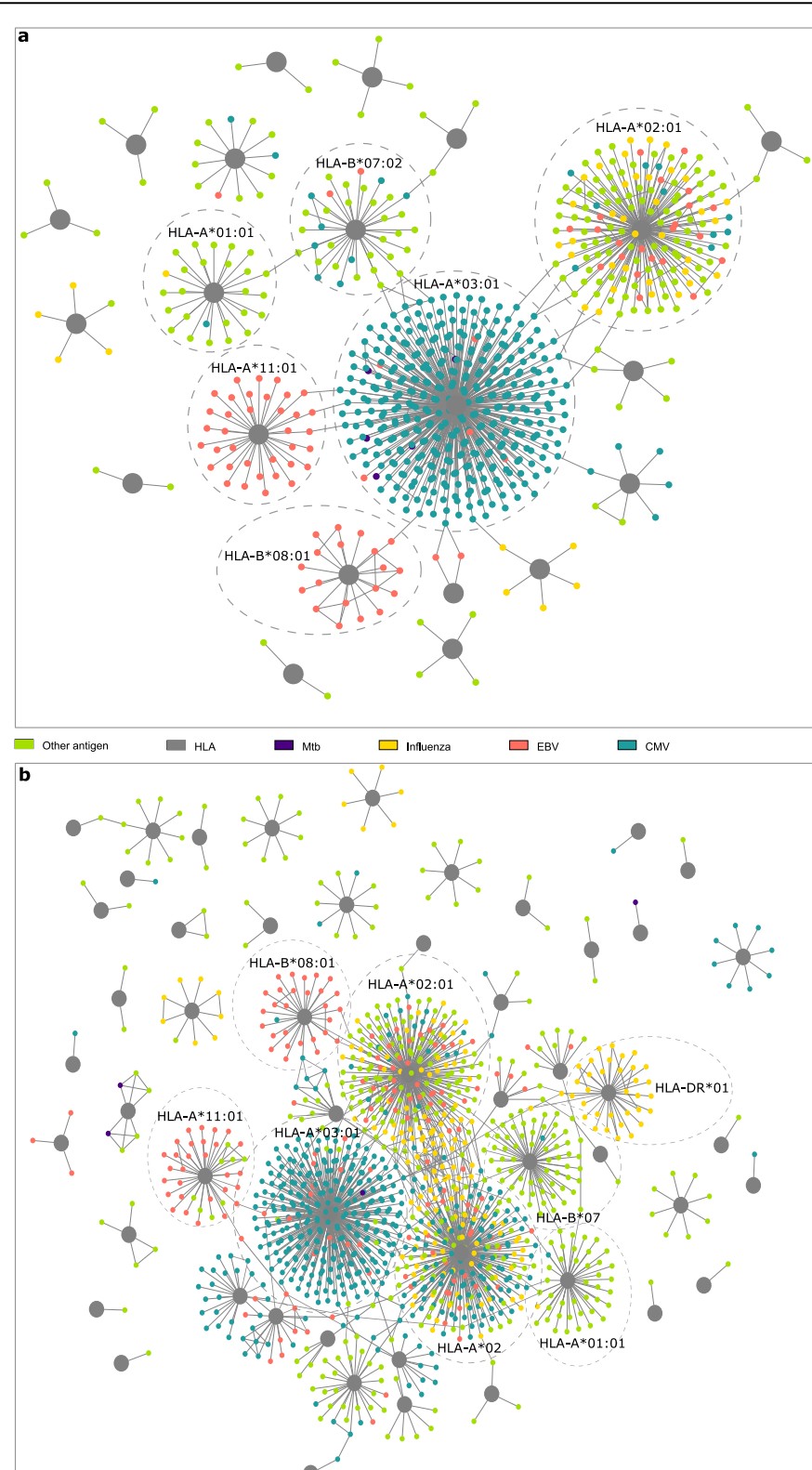

It should be mentioned that public databases such as McPAS[30] and VDJdb[31] are far from complete and are biased towards common viral pathogens. Therefore, more systematic efforts to decode the antigenic specificity of TCRs are needed to enrich the databases with the antigenic specificity of public T cell clonotypes. Different techniques can be used to achieve this aim, such as yeast-display[46], phage-display[47], and receptor–antigen pairing by targeted retroviruses[48].

In conclusion, our results indicate the utility of coupling machine learning with paired population-level profiling of TCR repertoires and HLA genotypes in terms of identifying sets of public clonotypes that are restricted to common HLA alleles. Having the presenting HLA candidates, for a set of TCRs-of-interest, *e.g.* because these TCRs were identified as disease-specific, opens new experimental avenues. Although we illustrated here that these clonotypes can be used to infer the HLA background of a

sample, they are also markers of previous infections. By decoding their antigenic specificities, a better understanding of immune responses to infections can be attained at both the individual and population levels.

## Methods

### Ethical approval and sample collection

The study has been approved by the ethical committee at Kiel University under the following ethical votes: D441/16, D474/12, A161/08, A103/14, and A148/14. For the Norwegian cohort, namely, H3, CD4, and UC4 datasets were derived from the IBSEN III study which was approved by the South-Eastern Regional Committee for Medical and Health Research Ethics (Ref 2015/946-3) and performed in accordance with the Declaration of Helsinki. The USA-based samples are derived from the SPARC IBD cohort from the IBD Plexus research program maintained by the Crohn's & Colitis Foundation and described by Raffals et al.[49]. A written informed consent was collected from all participants prior to the beginning of the study and all ethical regulations relevant to human research participants were followed.

### T cell repertoire profiling

Two technologies were used for profiling the T cell repertoire, namely, the ImmunoSEQ assay (Adaptive Biotechnologies)[50,51] and the αβ TCR profiling assay (MiLaboratories). For the ImmunoSEQ assay, DNA was extracted from peripheral blood using Qiagen DNeasy Blood Extraction Kit, after that, V(D)J recombination sequences encoding either the TRA or the TRB repertoire were amplified using a bias-controlled PCR[50]. Then, the amplified PCR products were sequenced using high-throughput short-reads sequencing and dedicated software was used for constructing clonotypes from the generated sequencing reads. For the αβ TCR profiling assay (MiLaboratories), 300 ng of RNA isolated from PAXgene Blood RNA tubes were used to profile the T cell repertoire of peripheral blood. After library preparation, sequencing was conducted on a NovaSeq X with an average of 5 million reads per chain per sample. Clonotypes were assembled from the sequencing data using MiXCR[52]. For both TCR-Seq technologies, we started processing the samples by filtering non-productive clonotypes, e.g. clonotypes that contain a stop codon, or a frameshift mutation. After that, different V(D)J rearrangements with the same V and J genes as well as CDR3 amino acid sequences were collapsed into a single clonotype. Hence, the number of clonotypes is synonymous with the number of unique functional V(D)J recombination events.

### Genotyping and HLA imputation

All samples were genotyped using Illumina Infinium Global Screening Array bead chips (GSAMD-24v1-0_20011747_A1 or GSAMD-24v2-0_20024620_A1). Genotype quality control (QC) was conducted independently for each cohort, following the procedures outlined in the BIGWAS quality control pipeline[53]. Specifically, the pipeline performs chip type detection and normalizes variant names. Variants with a missingness rate exceeding 0.02 were excluded from further analysis. Additionally, variants are filtered by Hardy-Weinberg equilibrium with a threshold of $p = 10^{-5}$. Sample QC was disabled in the BIGWAS QC with the parameter '--skip-sampleqc=1'. To facilitate population structure analysis, each cohort was combined with 2504 samples from the 1000 Genomes Project for principal component analysis (PCA). Samples were projected onto the first two principal components, and those falling within the median ± 5 times the interquartile range (IQR) along these components were identified as Europeans.

HLA imputation was performed with the HLApipePublic pipeline available at https://github.com/ikmb/HLApipePublic. The pipeline filters the input dataset to variants within the region of 29 mb and 34 mb on chromosome 6 and aligns them to the imputation reference. Imputation of HLA alleles was calculated using the HIBAG algorithm[54], which applies attribute bagging (BAGging) to enhance prediction accuracy. The analysis utilized the multi-ethnic IKMB reference panel, as described in[23]. This approach generates predictions by averaging HLA-type posterior

probabilities across an ensemble of classifiers built on bootstrap samples[54]. The phasing of HLA alleles was performed with the tool SHAPEIT2[55]. Imputation was performed at full four-digit resolution for the following loci: *HLA-A*, *HLA-C*, *HLA-B*, *HLA-DRB3*, *HLA-DRB5*, *HLA-DRB4*, *HLA-DRB1*, *HLA-DQA1*, *HLA-DQB1*, *HLA-DPA1*, and *HLA-DPB1*.

### Identifying HLA allele-associated clonotypes

We followed the same framework developed by Emerson et al.[10] to identify HLA-associated clonotypes. For each allele in each of the six classical HLA loci, namely, *HLA-A*, *HLA-B*, *HLA-C*, *HLA-DR*, *HLA-DQ* and *HLA-DP* we binned samples into two categories, carriers and non-carriers. Subsequently, we compared the incidence of each public clonotype, defined as a clonotype detected in two individuals or more, in carriers and non-carriers. Thus, for each clonotype and HLA allele we obtained a $2 \times 2$ contingency table that contains the number of individuals that have this allele and this clonotype, have this clonotype but not the allele, have the allele but not the clonotype, or do not have the allele or the clonotype. After building the table, we used a one-sided Fisher's exact test to investigate the statistical association between public clonotypes and HLA alleles. Although Zahid et al.[13] optimized the significance cutoff for different alleles, due to the computationally intensive nature of this approach we used a fixed cutoff of $1 \times 10^{-4}$ to define clonotypes that are associated with a particular HLA allele.

### Developing the TCR-based HLA imputation framework

We used a similar approach to Zahid et al.[13] where we started by resolving the promiscuous association problem, where a single clonotype is associated to multiple HLA alleles. Although we observed this problem among alleles of the same gene, it was more apparent among alleles of different genes. A potential cause for this problem is linkage disequilibrium (LD) where due to the statistical association nature of the study, we identify clonotypes that are associated with a particular haplotype instead of a particular HLA allele. To resolve this, we followed the L1-regularized linear regression (L1LR) framework proposed by Zahid et al.[13] which was implemented in TensorFlow to increase computational performance and is summarized in the following set of equations.

For each clonotype c, we let $L(c)$ denote the number of HLA alleles associated with c, for clonotypes with $L(c) > 1$, we conducted an L1LR analysis by building two matrices for each clonotype:

1. The label $Y \in Nx1$ matrix, where N represents the number of samples, and the elements in Y are defined as follows

$$Y_i = \begin{cases} 1, & \text{if clonotype c is detected in the repertoire of the i}^{th}\text{sample} \\ 0, & otherwise \end{cases}$$

2. The input features matrix $X \in R^{N \times (L(c)+1)}$, where N represents the number of samples and the first L(c) columns indicating if the $i^{th}$ individual carries the $j^{th}$ HLA allele associated with c. The last dimension of the matrix X represents the normalized number of clonotypes defined as follows:

$$X_{i,Lc+1} = \sqrt{C(R_i)}$$

Where $C(R_i)$ is the total number of clonotypes in the $i^{th}$ repertoire.

After constructing these X and Y matrices, we used a logistic regression function to predict the presence of clonotype c in the $i^{th}$ repertoire from the feature matrix X as follows:

$$Y_i^{pred} = \frac{1}{1 + e^{-(XW+b)}}$$

Where W represents the learned weights and b the biases of the model, these parameters were learned using gradient descent with the following loss

function

$$Loss(W, b) = -\frac{1}{n}\sum_{i=1}^{n}[y_i \log\left(Y_i^{pred}\right) + (1 - y_i)\log\left(1 - Y_i^{pred}\right)] + \lambda\sum_{j=1}^{d-1}|w_j|$$

Where $\lambda$ is a non-learnable parameter, *i.e.*, it is not learned using gradient descent but has a fixed value. This L1 loss term, $\sum_{j=1}^{d-1}|W_j|$, is used to enforce sparsity. We kept training the models until all the $w_j < 1x10^{-4}$ except one, this remaining $w_j > 1 \times 10^{-4}$ being the weight of the allele this clonotype is mostly associated with. For a given $\lambda$, we ran a gradient descent-based optimization for a 1000 epochs, if we could not resolve the HLA association after these 1000 epochs, that is, there is more than one weight $> 1 \times 10^{-4}$, then the training loop is restarted with a new value for $\lambda$. Ten values for $\lambda$ were tried, ranging from 0.01 to 10. If we could not achieve the required optimization task of getting all the weights to be $< 1 \times 10^{-4}$ except one with these ten different $\lambda$ values, then we label the input clonotype as "unsolvable". This implies that this clonotype is strongly linked with a specific HLA haplotype, and with the current dataset in terms of sample size and population structure we cannot resolve the association to a specific allele within the associated haplotype. These clonotypes are filtered from all downstream analyses .

Subsequently, we arrange the dataset into HLA alleles with their associated clonotypes. For each clonotype in this set, we calculated a weight that indicates the strength of its association with its cognate HLA allele following the same approach described by Zahid et al.[13]. Specifically, we built two matrices for each clonotype $c$ which is associated with an allele $a$

1. the input matrix $X \in R^{Nx2}$ where N represents the number of samples and $X_i = [C(c, R_i), \sqrt{C(R_i)}]$, where $C(R_i)$ returns the total number of clonotypes in the i^th sample and the $C(c, R_i)$ represents the expansion of clonotype c in the i^th repertoire. Subsequently, we normalized the matrix X using Z-score normalization applied column-wise.

2. the label matrix $Y \in R^{Nx1}$ which is defined according to the following indicator function

$$Y_i = \begin{cases} 1, & \text{if Subject}_i \text{ carries allele } a \\ 0, & \text{otherwise} \end{cases}$$

After that, we train a logistic regression model to predict the carriership of the allele $a$ from the expansion of clonotype $c$ in a repertoire $R$ profiled at a given depth. The absolute value of the $\beta_1$ weight of this model is the weight of clonotype $c$ that is associated with the HLA allele $a$.

These weights were used to calculate a weighted sum of allele-associated clonotypes' expansion in each repertoire. Lastly, a logistic regression classifier was developed to predict carriers and non-carriers of a given allele based on this weighted expansion. Specifically, we build two matrices to train these models:

1. the input matrix $X \in R^{Nx2}$ where N represents the number of samples and $X_i = [\sum_{j=1}^{A}|a_j| * C(c_j, R_i), C(R_i)]$ were the $\sum_{j=1}^{|A|}|a_j| * C(c_j, R_i)$ represents a weighted sum of the expansion of allele-associated clonotypes A. This set is made of tuples, $\{(a_j, c_j), \ldots\}$ where $a_j$ represents the weight of the j^th clonotype that is associated with a given allele and $c_j$ is the j^th clonotype that is associated with the same allele. The term, $C(c_j, R_i)$, represents the expansion of the j^th clonotype in the i^th repertoire. The $|a_j|$ term represents the absolute value of the clonotype weight identified by logistic regression as shown above, because we focused on the strength of association independent of its direction. Hence, if an allele a is associated with three clonotypes, $c_1, c_2$ and $c_3$ its set of associated clonotypes will be arranged as follow: $\{(w_1, c_1), (w_2, c_2), (w_3, c_3)\}$, where $w$ is the weight and $c$ is the clonotype. To calculate the weighted expansion of the allele associated clonotypes in the i^th repertoire, we can used the expanded form of the

equation described above as follows: $|w_1| * \text{expansion of } c_1 \text{ in } R_i + |w_2| * \text{expansion of } c_2 \text{ in } R_i + |w_3| * \text{expansion of } c_3 \text{ in } R_i$. Lastly, $C(R_i)$ is the total number of unique clonotypes present in the i^th repertoire.

2. the label matrix $Y \in R^{Nx1}$ defined according to the following indicator function

$$Y_i = \begin{cases} 1, & \text{if Subject}_i \text{ carries allele } a \\ 0, & \text{otherwise} \end{cases}$$

After that, we train a standard logistic regression model to predict the carriership of the allele $a$ in a repertoire $R$ profiled at a given depth using the allele-associated clonotypes set $A$ defined above.

## Clustering of HLA-associated clonotypes

To cluster HLA-associated clonotypes, we followed a graph-based approach in which each clonotype and HLA allele is defined as a node. To draw an edge between two nodes, these two nodes need to have the same V and J genes, in addition to having at most, one hamming distance between the CDR3 of their amino acid sequences. Alternatively, an edge can be drawn between two nodes if they represent the restriction of a specific clonotype toward an HLA molecule.

## Statistics and reproducibility

To identify TRA or TRB clonotypes associated with a given HLA allele, we used a one-sided Fisher's exact test to compare the prevalence of each public clonotype in carriers and non-carriers of this HLA allele. Three metrics were used to evaluate the performance of the model on the validation and the testing datasets, namely, balanced accuracy, precision, and recall.

## Reporting summary

Further information on research design is available in the Nature Portfolio Reporting Summary linked to this article.

## Data availability

Due to GDPR and consent restrictions, the datasets reported in the current study, namely, the T cell repertoires and paired HLA alleles of German individuals, can be obtained by submitting a project application to the popgen 2.0 Network (https://portal.popgen.de/). Regarding the Norwegian samples, upon contact with Prof. Marte Lie Høivik (m.l.hoivik@medisin.uio.no) an institutional data transfer agreement can be established and data shared if the aims of data use are covered by ethical approval and patient consent. The procedure will involve an update to the ethical approval as well as review by legal departments at both institutions, and the process will typically take one to two months from initial contact. Lastly, the US-based datasets are available upon approved application to the Crohn's & Colitis Foundation IBD Plexus Program (https://www.crohnscolitisfoundation.org/ibd-plexus).

## Code availability

The developed imputation models and HLA-restriction databases are publicly available under an academically permissive license at the project GitHub page (https://github.com/ikmb/TCR2HLA) and at Zenodo[56]. Developing and construing these models was performed using widely used Python libraries including Pandas (Version: 2.2.2), SciPy (Version: 1.15.3), Scikit-learn (Version: 1.3.1) and NumPy (Version: 1.26.4) while visualization was performed using Matplotlib (Version: 3.9.2) and seaborn (Version: 0.13.2). The large dataset of TRA and TRB clonotypes assembled in the current study were processed using Spark deployed on high performance computing framework.

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

## Acknowledgements
We would also like to thank the Crohn's & Colitis Foundation IBD Plexus program for providing us with the T cell repertoire profiles and the genotypes of the SPARC IBD cohort. We would also like to thank Sören Franzenburg, Janina Fuß, Rebekka Kraemer, Nicole Braun, Maria Eloina Figuera Basso, Anja Tanck, Yewgenia Dolshanskaya, and Melanie Vollstedt for their help with T cell repertoire profiling. We would also like to thank Michel Wittig and Tanja Wesse for their help with SNP array genotyping and for Christoph Prieß and Lars Wienbrandt for providing computational support to the project. The project was funded by the EU Horizon Europe Program grant *miGut-Health: Personalized blueprint of intestinal health* (101095470). The project also received infrastructure support from the EU program for Research and Innovation 'Horizon Health' (HORIZON-HLTH-2023-DISEASE-03) ID-DarkMatter-NCD (897856542). Additionally, the project received funding from the German Research Foundation (DFG) Research Unit 5042: miTarget – The Microbiome as a Therapeutic Target in Inflammatory Bowel Diseases along with funding from the DFG Cluster of Excellence 2167 "Precision Medicine in Chronic Inflammation (PMI)" and the DFG project EL 831/7-1. A.K.H.M. is funded by the DFG collaborative research center CRC 1526 "Pathomechanisms of Antibody-mediated Autoimmunity (PANTAU) – Insights from Pemphigoid Diseases". The SPARC IBD cohort is maintained by the Crohn's & Colitis Foundation for research use.

## Author contributions
H.E., A.F., and A.D. designed and conceived the study. A.K.H.M. profiled the TRA and TRB repertoires. E.M.W., M.G., and D.E. performed HLA imputation from genotyping arrays. H.E. integrated the datasets and developed the imputation models, and constructed the database containing TCRs with their restricted HLA allele. H.E. and A.K.H.M. wrote the first draft of the manuscript with subsequent contributions from all co-authors.

## Funding

## Competing interests
The authors declare no competing interests.
