## [Transparent Peer Review file · Communications Biology]

TCR Clonotypes as Predictors of HLA Carriership and Antigenic Exposure Histories

Corresponding Author: Dr Hesham ElAbd

Version 0:

Reviewer comments:

Reviewer #1

(Remarks to the Author)

ElAbd et al. present models for imputing HLAs from the TCR alpha and TCR beta repertoires, respectively. They demonstrate that public TCRs present in the repertoire may be associated to specific HLAs and subsequently use this association to build performant predictive models of HLAs. The methodology is straightforward and well suited to the problem. In particular, disentangling linkage disequilibrium is an important challenge which is well-addressed by the methodological choices. The authors acknowledge that the specific approach is not new but the dataset is the largest of its kind and the novel aspect of their work is making the data and tools publicly available. These data are extremely valuable and making them publicly available is an important contribution of this work warranting its publication.

I have a question that I would like clarified and few suggestions the authors may consider that I believe will improve the manuscript. I do not need to see a revised draft as these suggestions do not impact what I consider the most valuable new contribution of this paper, namely the data.

1. I was unable to ascertain what data will specifically be shared and in what format. The most valuable data would be the repertoires with HLA genotyping and the subsequent list of HLA associated TCRs. Hopefully this is what the authors intend to share. However, I understand that repertoire data are expensive to generate and the very least, I expect the authors to share the HLA associated sequences. While the repertoires would be of great value to the community for general applications, the HLA associated sequences are of particular value because they represent public TCRs that are likely turn up in many analyses and having a database of their HLA associations will provide interpretive power for other studies.

2. Figures in the supplement which showed the allele frequency vs number of associated sequences were unclear. There appear to be 4 figures all showing this but they were not labelled with enough detail to make clear which datasets they were representing. More importantly, I think the paper would benefit from discussion based on Figure S1. Please take a look at Figure 5 of DeWitt et al. 2018 (<https://elifesciences.org/articles/38358>). I think reproducing a figure like this in the main text would substantively contribute to the paper and help facilitate discussion around this topic which is already presented in the manuscript.

3. The treatment of cis and trans class II heterodimers could be made more clear. This is a really important issue that many researchers working with HLA data often ignore or do not fully understand. I think it's worth expanding the discussion a bit on this topic. Tolleson et al. (2012, <https://pmc.ncbi.nlm.nih.gov/articles/PMC3340161/>) point out the instability of some trans-complemented HLA-DQ heterodimers and provide general pairings rules. We expanded on these rules in our HLA analysis (<https://www.biorxiv.org/content/10.1101/2024.04.01.587617v2.full>) by deriving rules for HLA-DP. I think a more rigorous comparison of these studies is worthwhile. In particular, in Zahid et al. we provide a supplementary data table (Table 3) in which all cis-complemented class HLAs are inferred by the fact that they alpha and beta chains are in linkage disequilibrium. Conversely, trans-complemented HLAs are inferred as those being in equilibrium. I think it would be benefit the manuscript if the authors engage a bit more substantively with this important issue.

4. One of the most exciting aspects of this type of work is the fact that it begins to decode the immunogenetics and exposure history of subjects. The authors discuss this issue but I think it's a bit subtle and may not be fully appreciated by many readers. Specifically, if you use a case/control setup for repertoires with disease as your label performing a similar analysis to what is done in this study, what you get is a set of public TCRs enriched for specificity to the disease. However, the TCRs

will be responding to different antigens in different HLA contexts and will only be observed in subjects who have the HLA and disease exposure. Conversely, if HLA is the label, what you get is a set of TCRs enriched for specificity to the HLA. However, now these TCRs are responding to antigens derived from common exposures and are only present in subjects who have the HLA + exposure. Thus, the immune exposure history is encoded in the repertoire. This is not obvious to most readers and I think the authors may take the opportunity to communicate this more clearly since it is really cool that this is possible.

I believe this is a fine manuscript providing important value to the community. I applaud the authors effort to make this important area of research more accessible to the broader community of researchers.

Jabran Zahid

Reviewer #2

(Remarks to the Author)
Dear editor and authors,

Thank you for giving me the opportunity to review this manuscript. I must start with the caveat that this is not my main research field, but nevertheless I found the work very interesting. The authors compiled a large dataset of TCR clonotypes, paired with HLA haplotypes, and used this to train regression models that allow for imputation of HLA alleles from clonotype sets alone. Interestingly, they compared the performance of the models across HLA loci and also included models for inferring HLA carriership from TRA sequences. The latter models do not perform as well as the TRB-trained models (which are trained with a much larger dataset), but it is still interesting to try this. The authors claim that their dataset and method are publicly available.

1. Major Comments

1.1 This leads me to my first major issue with the work: As far as I can tell, the authors have not yet published their data and code. As this is one of the major selling points of the paper, I found this a bit disappointing.

More importantly, the authors refer to 3 similar studies, and claim that these teams have not published their data or their tools. This is not accurate: The Emerson et al data set is available here: <https://clients.adaptivebiotech.com/pub/Emerson-2017-NatGen> and Ortega et al (now published on Plos Comp Bio) has a tool on github and PyPI: <https://github.com/statbiophys/HLAGuessr>

Therefore, the authors have to (a) change the justification of their paper: other publicly available tools do exist, and (b) compare their method with the previously published methods. As the authors suggested, the details of the training dataset are quite important for the tool (e.g. disease-associated clonotypes, sample size), and therefore it would be really interesting to see how these tools compare across different datasets (even if the new tool is not "better" than previously-published methods).

1.2 The current methodology predicts HLA carriership on a per-allele basis. If we want to use this tool in practice, we would probably calculate log-odds for a long list of alleles, and then call the allele with the highest log-odds. It would therefore be more relevant to see how such an ensemble HLA-allele prediction model performs in comparison to the current per-allele approach. My reasoning is as follows: There are indeed thousands of HLA alleles discovered, but many of these alleles are very similar. Hence, I expect that similar alleles have similar log-odds, but perhaps the true allele will get the highest log-odds assigned. And therefore this ensemble model would be able to call the correct allele, even if the single-allele-model gives a lot of false-positives.

An obvious complication here is that for each locus, there are two alleles (for heterozygous individuals), but perhaps there is a way to solve this (e.g. call the 2 highest log-odds values).

2. Minor comments

- Increase font size in figures.

- line 57-60: The description of the Emerson et al study is not completely clear to me: what are "matched HLA-A and HLA-B allele calls"?

- line 72: The Emerson et al data set is available here: <https://clients.adaptivebiotech.com/pub/Emerson-2017-NatGen> Also, do you mean "publicly available tool?" instead of "dataset"?

- Line 70: "neither did they provide a detailed investigation of the factors shaping the predictive performance of these models": Is this one of the objectives of this paper? In that case, make that clear. The other authors performed a lot of validation and testing steps as well, so I would not say that such a detailed investigation is missing.

- line 77-78: this sentence is a bit awkward. Please rephrase. Also: together with the dataset, this is the main point of the paper, so elaborate a bit more.

- Can you explain what the benefit of imputing HLA genotype from TCR clonotypes? Why not use SNP? In what situation would this be useful? Also given that these imputations are not nearly 100% accurate.

- Is the performance of the model just a function of the size of the training set? e.g. TRA vs TRB: what happens if you down-sample TRB? Do you get similar results as for TRA, or is TRA "difficult" in a more fundamental way.
- About your datasets: are these published elsewhere? Or generated for this publication? This is not that clear.
- Line 118: How are you using the repertoire depth in the linear regression model? Please give more details of the model.
- Line 119: "by the running a given..." check sentence.
- Line 142-143: Why was resolving the HLA association not possible? From what I understand, the method adjusts the lasso penalty until there is exactly one HLA allele associated with the clonotype. In which situations is this not possible?
- Line 174: Is this Fig 2A? (not 1A)
- Figure 1B: "wait allele associations" should probably be "weight allele associations"
- Figure 1C: what do you mean by "cumulative expansion of allele-associated clonotype set"? What is expansion and why is it cumulative?
- Figure 2: is the test dataset the same as the validation dataset? Sometimes "test" and "validation" have different functions. Perhaps refer to the 20% left-out data as the "test" dataset, and call the Rosati and immunoCODE datasets the "validation" data.
- Figs 3, 4, 5: make sure the x-axes have the same limits (0-1)? so that we can better compare the method across datasets
- Add a table/figure showing the overall performance of each model w.r.t. each dataset (incl the 20% test set), as it is hard to read the overall performance from figs 2-5. (see also Issue 1.2)
- Line 332: "it still represents only of the TRB-HLA dataset": rephrase.
- Highlight the novelty of TRA-HLA models in the introduction? Even if it does not perform so well? Perhaps that is an interesting result by it self?
- Does the full model use some kind of regularization? Other authors have included this. Is the model over-fitting? How do the predictions for observations in the training dataset compare to predictions in the test dataset?
- Line 603: "disease associated clonotypes" should this be "HLA-associated"?
- Can you comment on the choice of the FET? One problematic assumption of this test is that the marginals of the contingency table are fixed. Is this a valid assumption in your case?
- Line 615-632: Please give more details. Your method appears to be a bit different from the one in Zahid et al, so it would help to have e.g. equations that define the L1-model and the logistic classifier. Also, The methods in Zahid et al are not very clear (and I don't think their Eq (1) is correct. See <https://www.biorxiv.org/content/10.1101/2024.04.01.587617v2.full>).
- Fig S1: Typo: "Pearson correlation co-efficiency"
- Fig S3: what does the colormap indicate? Please remove. (also in other SI figures)

Version 1:

Reviewer comments:

Reviewer #1

(Remarks to the Author)

The authors have addressed all my concerns. I'm happy to enthusiastically recommend this article for publication.

Reviewer #2

(Remarks to the Author)

Dear editor and authors,

Thank you for responding to all my concerns. I think in general the manuscript has improved, but I still have a couple of

issues. Most notably that many of the new edits would benefit from correcting grammar, spelling and overall structure. Especially the new methods section is far below publication quality.

Also, I don't completely agree with how the Ortega et al issue was handled: The authors now write: "However, all mentioned studies with the exception of Ortega et al., did not release a publicly available tool for imputing HLA alleles from TCR repertoires". My question is simply: why should we use your tool instead of Ortega et al? (the error you got while running their method can be easily resolved. See my comments below).

Here are some specific points:

Intro: Thank you for adding this final paragraph, and explaining the importance of your work. However, this paragraph could be better integrated with the rest of the introduction.

Page 6: "however, for a small subset of clonotypes this was not possible". Thank you for explaining this to me. However, I do think that adding a very brief explanation in the text will be helpful.

You could mention when the algorithm fails in the Online methods section and in the text say: "however, for a small subset of clonotypes this was not possible (see Online Methods)"

Figure 3: maybe you can remove the locus from the HLA type in the annotations? (it is a bit crowded) so write 03:03 instead of DQA1*03:03

page 16: "Although the assembled TRA-HLA dataset reported here represents, to the best of our knowledge, the biggest dataset of its kind, it still represents only (<25%) of the TRB-HLA dataset". I'm sorry, but this sentence still does not make sense. Maybe write:

"Although the assembled TRA-HLA dataset reported here is (to the best of our knowledge) the largest dataset of its kind, it is still relatively small (<25%) compared to the TRB-HLA dataset."

Online Methods; Developing the TCR-based HLA imputation framework:

Thank you for explaining the method in detail. I think it would help to check this text for redundancy and grammar, though.

For instance, the phrase is "as follows" (not "as follow"). Also, instead of writing

"For each clonotype c we have the function L which returns the number of alleles associated with clonotype c , hence, $L(c)$ return an integer that represent the number of HLA alleles that were associated with c ."

you can simply say "For each clonotype c , we let $L(c)$ denote the number of HLA alleles associated with c ."

and instead of "Following this we can construct the indicator function S , defined as follow: $S(c) = \dots$

For cases where $S(c) = 1$, we conducted an L1LR analysis..." you can just write:

"For clonotypes c with $L(c) > 1$ we conducted an L1LR analysis..."

There's no need to introduce an indicator function S .

There are a lot of needlessly convoluted sentences in this section, so please read through this part again and simplify.

page 23; point 2: What is S_i ? Subject i perhaps? Please clarify (or just say "if subject i carries allele a). Also: what is $f(x)$? Maybe remove this? Again, check spelling/grammar: "logistic region model".

Especially the last paragraph of page 23 is rather incomprehensible to me. Please fix this.

For example, why do you take the absolute value of the weight $|a|$?

Running HLAGuessr: The error is not raised if you add the option "-d tab" in the command line. I've added a comment to the issue on GitHub, with a suggestion how to solve the bug.

Figure 1C: "cumulative expansion of allele-associated clonotype set"

Thanks for explaining this. However, my suggestion was to explain this in the actual figure caption. In the caption, you are referring to a "weighted sum" instead (which is more accurate IMO).

Point 1.2: I've tried your model on the provided data, and often you get e.g. 2 HLA-A alleles with high probability. This makes sense for heterozygous individuals.

Therefore, I don't understand your response 2: "To provide a normalization form for the calculated P-value...". You're using softmax on the probabilities from a single locus. If someone has alleles $A^*01:01$ and $A^*01:02$, and the model correctly calls those with high probability (e.g. 0.98 and 0.99, and 0.00001 for the others), then the softmax would result in approx $p = 0.5$ for $A^*01:01$ and $p = 0.5$ for $A^*02:01$. That does not make sense to me. Are you using this normalization for the results in the paper? If so, I don't think you're explaining this.

Point-by-point response letter for the manuscript: “Decoding the restriction of T cell receptors to human leukocyte antigen alleles using machine learning” by EIAbd et al.

Editorial comments

All reviewers acknowledge the value of your dataset to the field. However, several specific concerns have been raised and need to be addressed. Note that, at a minimum, for us to send a further revised manuscript back to the reviewers, the datasets and code must be publicly available and should be generated specifically for this study, not published in another study.

We sincerely thank the editors and reviewers for acknowledging the strength of our study and its contribution to the field. We fully appreciate the importance of transparency and reproducibility, and we have taken concrete steps to ensure the availability of both data and code in alignment with the journal's policies.

Specifically, we are making the following data publicly available:

1. **TCR α (TRA) and β (TRB) clonotypes associated with common HLA alleles**
2. **Pre-trained models developed to impute HLA types based on either the TRA or the TRB repertoire**

These resources, and associated software, are accessible through the following repository: (<https://github.com/ikmb/TCR2HLA>).

Regarding the TCR repertoires and HLA genotyping data used in this study: access to these datasets is governed by ethical and legal constraints due to the nature of human subject data. While we cannot make them publicly available without restriction, we have provided a clear and detailed description in the manuscript (and in our responses to Reviewer I and Reviewer II) outlining the process by which qualified researchers may apply for access through the appropriate data access committees. This process complies with all relevant regulatory and ethical standards.

On the matter of dataset novelty: the **TRA and TRB clonotype databases linked to HLA alleles** represent a **novel dataset**, generated specifically for this study and not previously published. Similarly, the **HLA imputation models** developed here, based on both TRA and TRB data, **are entirely new** and have not been described in prior work.

Although the underlying TCR repertoires have been used in past research on immune-mediated inflammatory diseases (e.g., inflammatory bowel disease, primary sclerosing cholangitis), they

have **not** been analyzed in the context of **HLA-TCR associations** prior to this study. This work represents a significant re-analysis and reinterpretation aimed at uncovering novel immunogenetic insights, using newly developed tools and methodologies.

We hope this satisfactorily addresses the editorial concerns, and we remain committed to ensuring that our work meets the highest standards of scientific integrity and accessibility.

Reviewer I

1. I was unable to ascertain what data will specifically be shared and in what format. The most valuable data would be the repertoires with HLA genotyping and the subsequent list of HLA associated TCRs. Hopefully this is what the authors intend to share. However, I understand that repertoire data are expensive to generate and the very least, I expect the authors to share the HLA associated sequences. While the repertoires would be of great value to the community for general applications, the HLA associated sequences are of particular value because they represent public TCRs that are likely turn up in many analyses and having a database of their HLA associations will provide interpretive power for other studies.

We totally agree with the reviewer comment. Indeed, we are going to release:

1. The TCR repertoire and HLA genotyping results,
2. The TRB and TRA clonotypes associated with different HLA proteins,
3. The models that can be used for imputing the HLA clonotypes.

Both the imputation models and the TRA and TRB clonotypes associated clonotypes are publicly available at (<https://github.com/ikmb/TCR2HLA>). However, the release of TCR repertoire and HLA genotypes is regulated by ethical approvals, nonetheless, we provided a detailed description of how these repertoires and the HLA genotypes of these individuals be obtained in the manuscript. Namely, we added the following text to the data and code availability section:

“The developed imputation models and TRB HLA-restriction databases are publicly available under an academically permissive license at the project GitHub page (<https://github.com/ikmb/TCR2HLA>). Due to GDPR and consent restrictions the datasets reported in the current study, namely, the T cell repertoires and paired HLA alleles of German individuals, can be obtained by submitting a project application to the popgen 2.0 Network (<https://portal.popgen.de/>). Regarding the Norwegian samples, upon contact with Marte Lie Høivik (m.l.hoivik@medisin.uio.no) an institutional data transfer agreement can be established and data shared if the aims of data use are covered by ethical approval and patient consent. The procedure will involve an update to the ethical approval as well as review by legal departments at both institutions, and the process will typically take one to two months from initial contact. Lastly, the US-based datasets are available upon approved application to the Crohn's & Colitis Foundation IBD Plexus Program (<https://www.crohnscolitisfoundation.org/ibd-plexus>).”

2. Figures in the supplement which showed the allele frequency vs number of associated sequences were unclear. There appear to be 4 figures all showing this but they were not labelled with enough detail to make clear which datasets they were representing. More importantly, I think the paper would benefit from discussion based on Figure S1. Please take a look at Figure 5 of DeWitt et al. 2018 (<https://elifesciences.org/articles/38358>). I think reproducing a figure like this in the main text would substantively contribute to the paper and help facilitate discussion around this topic which is already presented in the manuscript.

We would like to thank the Reviewer for his clear and well formulated comment; we have edited our Figure 2 accordingly as depicted below and added the following section to the manuscript to discuss our findings in the light of DeWitt *et al.* study and the revised figure:

Using 80% of the data we were able to identify 722,060 clonotypes that were associated with 312 HLA alleles with the number of associated clonotypes being a function of allele frequency (**Fig. 2a-f**). Within the HLA-A locus, only eight alleles had a carriership frequency above 5% with the most frequent HLA alleles being HLA-A*02:01 with a carriership frequency of ~50% followed by HLA-A*01:01 (**Fig. 2a**). Given the higher diversity at the HLA-B*08:01 alleles, there was 13 alleles with a carriership frequency above 5%, with HLA-B*08:01 and HLA-B*07:01 being the most frequent and also the ones with the highest number of associated clonotypes (**Fig. 2b**). A similar pattern was observed at the HLA-C locus with only, eleven alleles showing more than 5% carriership frequency where the HLA-C*07:01, HLA-C*07:02 and HLA-C*06:02 being the most common and the alleles with the highest number of associated clonotypes (**Fig. 2c**). Although the three HLA-I loci, showed the same positive correlation between allele frequency and the number of associated clonotypes was different among with them, with the HLA-B locus having the highest number of associated clonotypes followed by the HLA-A locus and lastly HLA-C locus (**Fig. 2a-2c**).

Within the HLA-II alleles, the HLA-DR locus had the highest number of associated clonotypes, with only nine alleles having a carriership frequency above 5% with the HLA-DRB1*07:01 and HLA-DRB1*15:01 being the two HLA-DR proteins with the highest number of associated clonotypes (**Fig. 2d**). Within the HLA-DQ proteins, which are generated from the pairing of proteins encoded by the HLA-DQA1 locus and the HLA-DQB1 locus, nine HLA-DQ dimmers have a carriership frequency above 5% (**Fig. 2e**). The most frequent HLA-DQ complex was derived from HLA-DQA1*01:02-DQB1*06:02, followed by the HLA-DQA1*01:01-DQB1*05:01 and the

HLA-DQA1*05:01-DQB1*03:01 proteins (Fig. 2e). Only three HLA-DP complexes had a carriership a higher frequency above 5% namely, HLA-DPA1*01:03-DPB1*02:01, followed by HLA-DPA1*01:03-DPB1*04:01 and lastly, HLA-DPA1*02:01-DPB1*04:01 (Fig. 2f).

Most of the clonotypes were restricted to HLA-II alleles (n=466,277), particularly, HLA-DRB1 (n=303,330) relative to all HLA-I alleles (n=145,224), potentially, because of the higher ratio of CD4⁺ T cells in the blood relative to CD8⁺ T cells. These findings also confirm previous reports by DeWitt *et al.*²⁴, specifically, the strong positive correlation between allele frequency and the number of associated clonotypes, the higher number of associated clonotypes with HLA-II alleles and lastly, the low number of clonotypes associated with HLA-C locus. Using the L1-regularized logistic regression framework¹¹ we were able to resolve the association between clonotypes and multiple HLA alleles, however, for a small subset of clonotypes this was not possible. Specifically, out of the 600,095 associated clonotypes, 587,224 (97.8%) clonotypes were associated with a single HLA allele while only 12,871 (2.2%) clonotypes were associated with multiple alleles.”

Figure 1: The relationship between HLA allele carriership frequency (shown on the x-axis) and the number of associated TRB clonotypes (shown on the y-axis) for the six classical HLA proteins. HLA-A (a), HLA-B (b), HLA-C (c), HLA-DR (d), HLA-DQ (e), HLA-DP (f). P.corr denotes the Pearson correlation coefficient. Panels g-l, the relationship between HLA-allele carriership frequency and the performance of its TRB-based imputation model on a test dataset of 1,111 TRB repertoires with linked HLA calls. The performance metrics were used to evaluate the model performance, namely, balanced accuracy, recall and precision. (g-l) depict the performance of three HLA-I proteins, namely, HLA-A, HLA-B and HLA-C, respectively. Similarly, the performance of HLA-II proteins is illustrated in (g-i), with HLA-DR shown in (j), HLA-DQ in (k) and lastly, HLA-DP in (l). The data supporting panels (g-l) are provided in table S1.”

We have also edited the following supplementary figures accordingly:

Figure S9: The relationship between HLA allele carrier frequency (shown on the x-axis) and the number of associated TRB clonotypes (shown on the y-axis) for the six classical HLA proteins. HLA-A (a), HLA-B (b), HLA-C (c), HLA-DR (d), HLA-DQ (e), HLA-DP (f). P.corr donates the Pearson correlation co-efficiency.

Figure S12: The relationship between HLA allele carriership frequency (shown on the x-axis) and the number of associated TRA clonotypes (shown on the y-axis) for the six classical HLA proteins. HLA-A (a), HLA-B (b), HLA-C (c), HLA-DR (d), HLA-DQ (e), HLA-DP (f). Lastly, P.corr donates the Pearson correlation co-efficiency.

Figure S19: The relationship between HLA allele carrier frequency (shown on the x-axis) and the number of associated TRA clonotypes (shown on the y-axis) for the six classical HLA proteins. HLA-A (a), HLA-B (b), HLA-C (c), HLA-DR (d), HLA-DQ (e), HLA-DP (f). Lastly, P.corr donates the Pearson correlation co-efficiency.

3. The treatment of cis and trans class II heterodimers could be made more clear. This is a really important issue that many researchers working with HLA data often ignore or do not fully understand. I think it's worth expanding the discussion a bit on this topic. Tollefsen et al. (2012, <https://pmc.ncbi.nlm.nih.gov/articles/PMC3340161/>) point out the instability of some trans-complemented HLA-DQ heterodimers and provide general pairings rules. We expanded on these rules in our HLA analysis (<https://www.biorxiv.org/content/10.1101/2024.04.01.587617v2.full>) by deriving rules for HLA-DP. I think a more rigorous comparison of these studies is worthwhile. In particular, in Zahid et al. we provide a supplementary data table (Table 3) in which all cis-complemented class II HLAs are inferred by the fact that they alpha and beta chains are in linkage disequilibrium. Conversely, trans-complemented HLAs are inferred as those being in equilibrium. I think it would be benefit the manuscript if the authors engage a bit more substantively with this important issue.

We thank this Reviewer for rising these very important points and we edited the manuscript as follows:

To investigate this, we inferred HLA-DQ and HLA-DP haplotype structures statistically (Fig. S7; Table S2 and S3) and compared the performance of cis and trans HLA-DP and HLA-DQ complexes (Fig. S8). Most of the alleles with a carrier frequency >1% were potentially cis

complexes and we had models for only three trans HLA-DQ complexes, namely, HLA-DQA1*03:01-DQB1*03:03, HLA-DQA1*02:01-DQB1*04:02, and HLA-DQA1*05:05-DQB1*02:02. Similarly, for HLA-DP complexes only one trans-complex had a frequency >1%, specifically, HLA-DPA1*01:03-DPB1*16:01, but all other HLA-DP complexes (n=15) were potentially cis complexes. Thus, our observations suggest that not all potential cis HLA-DP or HLA-DQ complexes can be accurately imputed from the TRB repertoires.

Figure S7: The identification of cis and trans HLA-DQ (a) and HLA-DP (b) complexes by comparing their observed HLA-DQ and HLA-DP frequencies to the expected frequency of carriership frequency from their individual's alpha and beta genes using a chi-square test. We used Bonferroni correction to correct for multiple testing where trans complexes are depicted in grey and potential cis complexes (chi-square P-value > Bonferroni-corrected threshold) in a color-gradient representing their association P-value in both panels.

Figure S8: The performance of cis and trans HLA-DQ and HLA-DP complexes models on the validation dataset. (a), (b) and (c) depict the balanced accuracy, recall and precision of cis and trans HLA-DQ complexes while (e), (f) and (g), show the performance of the same performance metrics across cis and trans HLA-DP complexes.”

4. One of the most exciting aspects of this type of work is the fact that it begins to decode the immunogenetics and exposure history of subjects. The authors discuss this issue but I think it's a bit subtle and may not be fully appreciated by many readers. Specifically, if you use a case/control setup for repertoires with disease as your label performing a similar analysis to what is done in this study, what you get is a set of public TCRs enriched for specificity to the disease. However, the TCRs will be responding to different antigens in different HLA contexts and will only be observed in subjects who have the HLA and disease exposure. Conversely, if HLA is the label, what you get is a set of TCRs enriched for specificity to the HLA. However, now these TCRs are responding to antigens derived from common exposures and are only present in subjects who have the HLA + exposure. Thus, the immune exposure history is encoded in the repertoire. This is not obvious to most readers and I think the authors may take the opportunity to communicate this more clearly since it is really cool that this is possible.

We would like to thank this Reviewer for the thoughtful comment. We edited our discussion to reflect this point as described below, with changes highlighted in the text below: “Although we could not resolve the antigenic specificity for all HLA-associated clonotypes using public databases^{28,29}, we were able to annotate the antigenic specificity for >1,000 TRA and TRB clonotypes. Most of these clonotypes were specific to common viral and bacterial infections.

Similar observations were observed by analysing the antibody repertoire of large cohorts using high throughput assays such as phage-immunoprecipitation sequencing (PhIP-Seq)³⁶⁻³⁸. In these studies, most public responses were against common viral and bacterial antigens such as EBV, CMV and other common bacterial infections³⁶⁻³⁸. Hence, these findings indicate that common bacterial and viral infections are a major determinant of the immune repertoire at both the cell mediated and humoral immune response. While the HLA influences humoral immune responses³⁹, its effect is much stronger at the cell-mediated immune responses because of HLA-restriction. Thus, the identified allele-associated clonotypes represent a fraction of the immune memory toward prevalent infections.

Mechanistically, the interaction between infections and HLA proteins will lead to the formation of different HLA-peptide complexes in different individuals, these complexes will be recognized by different T cells. Hence, carriers of a given allele will have a shared response in terms of T cell clonotypes toward the same HLA-peptide complex relative to individuals with different HLA alleles. Thus, the subset of allele-associated clonotypes present in an individual represent a fraction of the immune exposure history of this individual. By decoding the antigenic specificity of these clonotypes a better understanding of the immunological history of an individual and/or a population can be developed. Alterations in responses to some of these common infections have been associated with different diseases, for example, inappropriate immune responses toward Epstein-Barr virus have been associated with multiple autoimmune and chronic inflammatory diseases such as multiple sclerosis⁴⁰, rheumatoid arthritis⁴¹, and Sjögren syndrome⁴² and primary sclerosis cholangitis¹⁷. Additionally, cytomegalovirus has been linked with the adult-onset Still's disease⁴³, and human papillomaviruses with cervical cancers⁴⁴. By analysing the immune repertoire of affected individuals and appropriate controls, we can identify a group of clonotypes that are expanded in affected individuals. These clonotypes will be restricted to specific HLA alleles and recognize a prevalent antigenic exposure within the affected individuals¹⁶⁻¹⁸. Hence, by decoding the antigenic exposure of these disease-associated and HLA-restricted clonotypes we can decode the antigenic exposure trajectory involved in a particular disease.”

Reviewer II

1. Major Comments

1.1 This leads me to my first major issue with the work: As far as I can tell, the authors have not yet published their data and code. As this is one of the major selling points of the paper, I found this a bit disappointing.

We thank the reviewer for the comment, and we totally understand his concern; indeed, we are going to release:

1. The TCR repertoire and HLA genotyping results,
2. The TRB and TRA clonotypes associated with different HLA proteins,
3. The models that can be used for imputing the HLA clonotypes.

Both the imputation models and the TRA and TRB clonotypes associated clonotypes are publicly available at (<https://github.com/ikmb/TCR2HLA>). However, the release of TCR repertoire and HLA genotypes is regulated by ethical approvals, nonetheless, we provided a detailed description of how these repertoires and the HLA genotypes of these individuals be obtained in the manuscript. Namely, we added the following text to the data and code availability section:

“The developed imputation models and TRB HLA-restriction databases are publicly available under an academically permissive license at the project GitHub page (<https://github.com/ikmb/TCR2HLA>). Due to GDPR and consent restrictions the datasets reported in the current study, namely, the T cell repertoires and paired HLA alleles of German individuals, can be obtained by submitting a project application to the popgen 2.0 Network (<https://portal.popgen.de/>). Regarding the Norwegian samples, upon contact with Marte Lie Høivik (m.l.hoivik@medisin.uio.no) an institutional data transfer agreement can be established and data shared if the aims of data use are covered by ethical approval and patient consent. The procedure will involve an update to the ethical approval as well as review by legal departments at both institutions, and the process will typically take one to two months from initial contact. Lastly, the US-based datasets are available upon approved application to the Crohn's & Colitis Foundation IBD Plexus Program (<https://www.crohnscolitisfoundation.org/ibd-plexus>).”

Regarding the second point of the comment: “More importantly, the authors refer to 3 similar studies and claim that these teams have not published their data or their tools. This is not accurate: The Emerson et al data set is available here: <https://clients.adaptivebiotech.com/pub/Emerson-2017-NatGen> and Ortega et al (now published on Plos Comp Bio) has a tool on github and PyPI: <https://github.com/statbiophys/HLAGuessr>” we thank this Reviewer for pointing us to the

published study by Ortega *et al.* which was still as a preprint at the time of writing and submitting the manuscript. Regarding the two other datasets we mentioned in the manuscript we would like to highlight two points

1. Emerson *et al.* have released the TRB repertoire and the HLA typing for the HLA-A and HLA-B loci only and did not release any model for HLA imputation.

3. The study by Zahid *et al.* is at the preprint stage and has not released any dataset. Thus, these two points confirm what we wrote in the introduction, however, we edited the introduction to highlight the Ortega *et al.* study further and draw a stronger contrast between our study and Ortega *et al.* study as depicted below:

This leads us to the third part of the part of the reviewer comment: “Therefore, the authors have to (a) change the justification of their paper: other publicly available tools do exist, and (b) compare their method with the previously published methods. As the authors suggested, the details of the training dataset are quite important for the tool (e.g. disease-associated clonotypes, sample size), and therefore it would be really interesting to see how these tools compare across different datasets (even if the new tool is not "better" than previously published methods).” Basically, the Reviewer’s argument can be separated into:

1. Commenting on the availability of data and tools, this point has been addressed early on where we mentioned that we are going to release: (i) code for using these models, the database containing clonotypes restricted to common HLA alleles and detailed the process for obtaining these clonotypes.

2. Comparing the developed tool to previously published tools. We tried to run the HLAGusser reported by Ortega *et al.* but we could not run the tool as it fails with the following error message “AttributeError: 'Values' object has no attribute 'infile_name'. Did you mean: 'outfile_name'?”. There is also an open issue at the tool GitHub page since April about the same issue but it has not been resolved until today (the 18th of July 2025; <https://github.com/statbiophys/HLAGuessr/issues/1>). Hence, we were not able to benchmark our models against the models developed by Ortega and colleagues. Further, the models reported by Zahid *et al.* are not publicly available. Hence, we were not able to benchmark our models against these models as well.

1.2 The current methodology predicts HLA carriership on a per-allele basis. If we want to use this tool in practice, we would probably calculate log-odds for a long list of alleles, and then call the allele with the highest log-odds. It would therefore be more relevant to see how such an ensemble HLA-allele prediction model performs in comparison to the current per-allele approach. My reasoning is as follows: There are indeed thousands of HLA alleles discovered, but many of these alleles are very similar. Hence, I expect that similar alleles have similar log-odds, but perhaps the true allele will get the highest log-odds assigned. And therefore this ensemble model would be able to call the correct allele, even if the single-allele-model gives a lot of false-positives. An obvious complication here is that for each locus, there are two alleles (for heterozygous individuals), but perhaps there is a way to solve this (e.g. call the 2 highest log-odds values).

We are not sure if we understand the Reviewer’s comment correctly. Currently our models are arranged as follow:

1. We have a collection of models, each of them predicts the carriership of a given allele, that is, the probability that an individual carries a given allele. The model does not take the carriership of other alleles into consideration. Because we have these probabilities, we can select the alleles with the highest probability score. From this probability we can calculate the log-odds ration using the following equation

$$\text{log Odds of carriership} = \ln\left(\frac{\text{Probability of carrier}}{1 - \text{Probability of carrier}}\right)$$

The log odds are a function of the probability of carriership and whether we choose the most likely allele per locus via the probability or the log odds, we will end up with the same one or two alleles.

2. To provide a normalization form for the calculated P-value across a locus, that is the set of alleles available at a specific locus such as HLA-A or HLA-B locus, we introduced a SoftMax function defined as follow

$$\text{normalized } P_{\text{allele } j} = \frac{e^{P_{\text{allele } j}}}{\sum_{i=1}^K e^{P_{\text{allele } i}}}$$

Where *normalized P_{allele j}* represent the SoftMax normalised allele carriership probability. This normalization gives an adjust probability distribution over all the support allele of a given locus. In both cases, we selected alleles with the highest P-value as the genotype of a given sample at a given HLA locus such as HLA-A or HLA-DRB1.

2. Minor comments

2.1. Increase font size in figures.

We thank the reviewer for his comments we have recreated all figures to increase the font and make it with the guidelines of Nature Communication Biology in terms of labelling and font size. Further, we added more annotations to provide more information to the readers.

2.2. line 57-60: The description of the Emerson *et al.* study is not completely clear to me: what are "matched HLA-A and HLA-B allele calls"?

We want to describe here that the dataset of Emerson *et al.* contained the TRB repertoire and HLA-A and HLA-B allotypes for 666 individuals. We have edited the sentence accordingly:

"One of the earliest studies was performed by Emerson and colleagues who used a statistical framework to analyze the TRB repertoire of 666 individuals with matched HLA-A and HLA-B allotypes, enabling them to identify thousands of TRB clonotypes that were restricted to multiple HLA-A and HLA-B alleles⁹."

2.3. The Emerson *et al.* dataset is available here: <https://clients.adaptivebiotech.com/pub/Emerson-2017-NatGen>. Also, do you mean "publicly available tool?" instead of "dataset"?

We thank the Reviewer for pointing us to the Emerson *et al.* dataset which we have used previously in other applications. Our sentences here were directed at the general problem that the other studies, except Ortegá *et al.*, have not (yet) released a tool that can be used for imputing HLA from either the TRA or the TRB repertoire. We have edited this sentence in the introduction accordingly: *"However, all mentioned studies with the exception of Ortegá *et al.*¹⁰, did not release a publicly available tool for imputing HLA alleles from TCR repertoires."*

2.4 - Line 70: "neither did they provide a detailed investigation of the factors shaping the predictive performance of these models": Is this one of the objectives of this paper? In that case, make that clear. The other authors performed a lot of validation and testing steps as well, so I would not say that such a detailed investigation is missing.

We acknowledge the Reviewer's perspective, and we have removed this phrase from the introduction which currently states: *"However, all mentioned studies with the exception of Ortegá *et al.*¹⁰, did not release a publicly available tool for imputing HLA alleles from TCR repertoires."*

2.5 - line 77-78: this sentence is a bit awkward. Please rephrase. Also: together with the dataset, this is the main point of the paper, so elaborate a bit more.

We edited the paper accordingly as follow “*After that we utilized the identified TRA and TRB clonotypes to develop machine-learning models that can be used to impute the carriership of common HLA alleles.*”

2.6 - Can you explain what the benefit of imputing HLA genotype from TCR clonotypes? Why not use SNP? In what situation would this be useful? Also given that these imputations are not nearly 100% accurate.

We have broken down the Reviewer’s comments into two replies:

1. The advantages of imputing HLA from the TRA or the TRB repertoire relative to SNP genotyping.

Imputing the HLA alleles from the T cell repertoire can reduce the cost of T cell repertoire analyses which are becoming increasingly common and are utilized to understand cancers, infection diseases and auto-immune diseases. In these analyses, obtaining the HLA allotypes of the analyzed samples is critical, for example, to resolve the HLA-restriction of disease-associated clonotypes. Hence, instead of performing HLA-typing and TCR repertoire profiling separately, by imputing the HLA carriership from the TCR-repertoire, there is no other costs needed to impute the HLA alleles. Beside cost-reduction, imputing HLA allotypes from TCRs enable us to identify functional HLA allotypes which is important in the typing of HLA-DQ and HLA-DP where theoretically four possible combinations are possible, however, some of these theoretical four combination are not functional.

2. Performance of these models relative to other methods such as genotyping?

While the performance of HLA imputation methods from genotyping data has improved over the years and became very accurate for common HLA alleles in Europe and the Northern America, its performance for rarer alleles is still far from a 100% accuracy. Further, the gold standard method for HLA-typing remains sequencing either Sanger-sequencing or Next-generation sequencing. Hence, each imputation methods is an approximation, nonetheless, because of the lower cost of performing imputation (and its amenability to high-throughput analysis and applicability to existing large-scale data sets) relative to gold-standard typing, it is still a commonly use approach for genetic studies.

We added the following section to the introduction to reflect on these points as follow: “As T cell repertoire analyses became an import method to understand tumors^{12,13}, response to infectious diseases^{14,15} and identifying antigen driving chronic inflammatory diseases^{16–18}, the developed imputation models can decrease the cost of these analyses by removing the need to perform laborious and expensive wetlab-based HLA-typing. Furthermore, these models can impute only functional HLA proteins shaping the repertoire which is especially important for HLA-DQ and HLA-DP proteins because both the α and the β chains are polymorphic however, not all assembled $\alpha\beta$ heterodimers produce a functional protein. Lastly, the generated datasets of TRA and TRB clonotypes associated with their cognate HLA allele, is of paramount importance for analyzing disease-associated clonotypes, which are a group of public clonotypes that are expanded in a specific disease¹⁶, for example, primary sclerosis cholangitis (PSC)¹⁷ or inflammatory bowel disease (IBD)^{18–20}.”

2.7 - Is the performance of the model just a function of the size of the training set? e.g. TRA vs TRB: what happens if you down-sample TRB? Do you get similar results as for TRA or is TRA "difficult" in a more fundamental way.

This is a very interesting point, however, a simple down-sampling in our case would not results in a fair comparison between the TRA and TRB repertoires because different technologies and biological materials were used to profile the repertoire. Specifically, TRB repertoires were profiled from DNA using the immunoSEQ assay, while the TRA repertoire were profiled from RNA using MiLaboratories kits. These technologies profile the repertoire at different depths with immunoSEQ generally enabling deeper repertoire profiling. Hence, a direct comparison between a down-sampled TRA and TRB would address variation introduced by differences in the sample size but would not address the variation or differences introduced by differences in the repertoire depth between the two technologies used to profile the TRA and TRB clonotypes. This is further complicated by differences in the cohorts from which the data was obtained. For example, the TRA repertoires were derived from Norway while TRB repertoires were derived from Germany and the United States. This difference in geographical origin and consequently allele variation can bias the comparison because of differences in allele frequency distribution between these two cohorts. Thus, it is not trivial to disentangle the differences between TRA and TRB clonotypes in their ability to impute HLA allele carriership.

In addition, we had two TRA datasets in the current manuscript. One was profiled from DNA but with a smaller sample size (n=385 samples) and a second larger dataset derived from RNA which had a bigger sample size (n=855). The second bigger dataset enabled the identification of a larger

number of TRA clonotypes that were restricted to common HLA alleles, and consequently, yielded a more accurate HLA imputation model. This indicates that sample size is the main driver behind the inferior performance of TRA relative to TRB, of course other biological reasons can contribute to some differences in imputing HLA alleles from the TRA or the TRB clonotypes, the effect of sample size will overrule these differences.

2.8 About your datasets: are these published elsewhere? Or generated for this publication? This is not that clear.

We assembled multiple TCR repertoire datasets that we previously generated. We have edited the manuscript by adding citations to Table 1 as depicted below and in the revised manuscript: “

Table 1: An overview of the datasets used for building HLA imputation models based on the T cell repertoire. HC refers to healthy controls, CD to individuals with Crohn’s disease and UC to individuals with ulcerative colitis. Lastly, symptomatic controls represent individuals with symptoms of inflammatory bowel disease, but their endoscopy and lab tests failed to unambiguously diagnose CD or UC.

DATASET NAME	LOCUS	NUMBER OF SAMPLES	COUNTRY	PHENOTYPE	TCR-SEQ METHOD
HC 1	TRB	773 ¹⁷	Germany	Healthy blood donors	Adaptive ImmunoSEQ
CD 1	TRB	1,186 ¹⁸	Germany	CD patients	Adaptive ImmunoSEQ
UC 1	TRB	480 ¹⁸	Germany	UC patients	Adaptive ImmunoSEQ
PSC 1	TRB	431 ¹⁷	Germany	PSC patients	Adaptive ImmunoSEQ
CD 2	TRB	1,809 ²⁰	USA	CD patients	Adaptive ImmunoSEQ
UC 2	TRB	854 ²⁰	USA	UC patients	Adaptive ImmunoSEQ
HC 2	TRA	165 ¹⁹	Germany	Healthy blood donors	Adaptive ImmunoSEQ
CD 3	TRA	124 ¹⁹	Germany	CD patients	Adaptive ImmunoSEQ
UC 3	TRA	96 ¹⁹	Germany	UC patients	Adaptive ImmunoSEQ
HC 3	TRA	264 ¹⁹	Norway	Symptomatic controls	MiLaboratories kits
CD 4	TRA	231 ¹⁹	Norway	CD patients	MiLaboratories kits
UC 4	TRA	360 ¹⁹	Norway	UC patients	MiLaboratories kits

2.9 Line 118: How are you using the repertoire depth in the linear regression model? Please give more details of the model.

Repertoire depth represents the total number of clonotypes that are present in a sample, hence, it is a single number that is provided together with the cumulative expansion of allele associated clonotypes to the final logistic regression models.

2.10 Line 119: "by the running a given..." check sentence.

We have revised the sentence as follow: "Lastly, we apply each of these allele-models to either the TRA or the TRB repertoire to calculate the HLA allele carriership of the provided repertoire (Fig. 1D)."

2.11 Line 142-143: Why was resolving the HLA association not possible? From what I understand, the method adjusts the lasso penalty until there is exactly one HLA allele associated with the clonotype. In which situations is this not possible?

Regarding the last point, resolving the HLA association was not possible in the case of strong linkage disequilibrium (LD). Regarding the first point about the lasso penalty, due current technologies and computational constrains, we need to run an L1LR model across hundreds of thousands of clonotypes, for each model we trained for a 1000 epochs with a preset selected parameter, λ , the value of this parameter was selected using a grid search. Nonetheless, this grid search needed to be bounded for the problem to be computationally traceable. Hence, in case an L1LR model was not able to resolve the HLA association for a specific clonotypes, using a bounded grid search, for example, because the optimal value of λ is outside the defined search range, we then are not able to resolve the HLA-restriction of a specific clonotypes. We have adjusted the method sections to reflect on this constrain as shown below:

"To resolve this, we followed the L1 regularized linear regression framework proposed by Zahid *et al.*¹³ which was implemented in TensorFlow to increase computational performance and summarized in the following set of equations.

For each clonotype c we have the function L which returns the number of alleles associated with clonotype c , hence, $L(c)$ return an integer that represent the number of HLA alleles that were associated with c . Following this we can construct the indicator function S , defined as follow:

$$S(c) = \begin{cases} 0, & \text{if } L(c) = 1 \\ 1, & \text{if } L(c) > 1 \end{cases}$$

For cases where $S(c) = 1$, we conducted an L1LR analysis by constructing the following matrices for each clonotypes:

1. The label Y vector which is $\in N \times 1$ where N represent the number of samples and elements in Y are defined as follow

$$Y_i = \begin{cases} 0, & \text{if } D(c, R_i) = 0 \\ 1, & \text{if } D(c, R_i) = 1 \end{cases}$$

Where the $D(c, R_i) = 1$ if clonotype c is detected in the repertoire of the i^{th} sample, otherwise the function returns 0.

2. The input features matrix $X \in R^{N \times (L(c)+1)}$ where N represent the number of samples and the first $L(c)$ columns are indicator columns that indicate if the i^{th} samples have the j^{th} element of the alleles associated with c . The last dimension of the matrix X represents the normalized number of clonotypes defined as follow:

$$X_{i,d+1} = \sqrt{C(R_i)}$$

Where $C(R_i)$ is a function that returns the total number of clonotypes in the i^{th} repertoire.

After constructing these X and Y matrices, we used a logistic regression function to predict the presence of clonotype c in the i^{th} repertoire from the set of associated alleles and normalized number of clonotypes defined in the feature matrix X using a logistic regression function defined as follow:

$$Y_i^{pred} = \frac{1}{1 + e^{-(XW+b)}}$$

Where W represent the weight matrix and b represents the biases of the models, the values of these learned parameters were learned using gradient decent with the following loss function

$$Loss(W, b) = -\frac{1}{n} \sum_{i=1}^n [y_i \log(Y_i^{pred}) + (1 - y_i) \log(1 - Y_i^{pred})] + \lambda \sum_{j=1}^{d-1} |w_j|$$

Where λ is a non-learnable parameter, that is, it is not optimized or learned during the gradient descent. This L1 loss term, $\sum_{j=1}^{d-1} |w_j|$, is used to force enforce sparsity. We keep training the model until all the $w_j < 1 \times 10^{-4}$ except one, where the remaining $w_j > 1 \times 10^{-4}$ being the weight of the allele this clonotypes is mostly associated with. For a given λ , we try running a gradient-decent based optimizer for 1000 epochs, if we could not resolve the HLA association after these 100 epochs, the training loop is restarted with a new value for λ . Ten values for λ were tried ranging from 0.01 to 10.

Subsequently, we arrange the dataset into HLA-alleles and their corresponding HLA associated clonotypes. For each clonotype in the set we can calculate a weight that can be interpreted as the strength of its association with its cognate allele following the same recipe described by Zahid *et al.*¹³. Specifically, we built two matrices for each clonotype c which is associated to an allele

a

1. the input matrix $X \in R^{N \times 2}$ where N represents the number of samples and $X_i = [C(c, R_i), \sqrt{C(R_i)}]$ where $C(R_i)$ represents the total number of clonotypes in the i^{th} sample and the $C(c, R_i)$ represents the expansion of clonotype c in the i^{th} repertoire. Subsequently, we normalized the matrix X using Z-score normalization applied column-wise.

2. the label matrix $Y \in R^{N \times 1}$ which is defined according to the following indicator function

$$Y_i = f(x) = \begin{cases} 1, & \text{if } S_i \text{ carries allele } a \\ 0, & \text{otherwise} \end{cases}$$

After that we train a standard logistic regression model to predict the carriership of the allele a from the expansion of clonotype c in a repertoire R profiled with a given depth. The β_1 weight of this logistic regression model is the weight of clonotype c with regard to predicting allele a .

These weights were used to calculate a weighted sum of the expansion of allele-associated clonotypes in each repertoire. Lastly, a logistic regression classifier to classify repertoires into allele carrier and non-carriers was developed. Specifically, we build two matrices to train logistic regression models to impute carriership alleles from the TRA or the TRB repertoire:

1. the input matrix $X \in R^{N \times 2}$ where N represents the number of samples and $X_i = [\sum_{j=1}^{|A|} |a_j| * C(c_j, R_i), C(R_i)]$ where the $\sum_{j=1}^{|A|} |a_j| * C(c_j, R_i)$ represents a weighted sum of the weighted allele associated clonotypes. A represents a set made of tuples, $\{(a_j, c_j), \dots\}$ where a_j represents the weight of the j^{th} clonotype that is associated with a given allele and c_j is the j^{th} clonotype that is associated with a given allele. The term, $C(c_j, R_i)$, represents the expansion or the count of the j^{th} clonotype in the i^{th} repertoire. Lastly, $|A|$ represents the number of elements in the set of clonotypes associated with a given allele.

2. the label matrix $Y \in R^{N \times 1}$ which is defined according to the following indicator function

$$Y_i = f(x) = \begin{cases} 1, & \text{if } S_i \text{ carries allele } a \\ 0, & \text{otherwise} \end{cases}$$

After that we train a standard logistic regression model to predict the carriership of the allele a from the expansion of clonotype c in a repertoire R profiled with a given depth.”

2.12 Line 174: Is this Fig 2A? (not 1A)

We edited the manuscript accordingly.

2.13 Figure 1B: "wait allele associations" should probably be "weight allele associations"

We updated the figure accordingly as depicted below and in the manuscript:

2.14 Figure 1C: what do you mean by "cumulative expansion of allele-associated clonotype set"? What is expansion and why is it cumulative?

1. What is expansion? Basically, the T cell repertoire is assembled from clones, each of these clones contain the same V(D)J combination. Hence, we can represent the structure of the repertoire in a tabular form, where each row is a unique V(D)J combination that is a clonotype, and then the number of times it has been observed that is clone size or the expansion of this clonotype.

2. What does “cumulative expansion” mean? After we discovered multiple clonotypes associated with each allele as depicted in Fig. 1B, we can depict the expansion of these clonotypes in a given repertoire by summing the expansion of these clonotypes, and hence we have a cumulative

expansion which is a single number that represents the fraction of clonotypes present in a given repertoire that is associated with a given allele.

2.15 Figure 2: is the test dataset the same as the validation dataset? Sometimes "test" and "validation" have different functions. Perhaps refer to the 20% left-out data as the "test" dataset, and call the Rosati and immunoCODE datasets the "validation" data.

Our definition for a validation dataset is a dataset that is used for finetuning the hyperparameters of the models and it is a fraction of the same dataset used for training, i.e. the dataset is split into a training (~80%) and a validation (~20%). However, the "test dataset" is an independent test that is derived from a different cohort or different technology to assess the performance of the model on real world data. Hence, we believe that our labelling of the different datasets is plausible and align with the established definitions of validation and test datasets.

2.16 Figs 3, 4, 5: make sure the x-axes have the same limits (0-1)? so that we can better compare the method across datasets

We thank the Reviewer for the recommendation, however, putting them on the same scale can compress variability among the different groups. However, to provide an easy to interpret visualization among the different figures we annotated each dot with the corresponding allele

name to make comparing the performance among the different test datasets easier for the readers as depicted below.

Figure 2: The performance of the TRB-based HLA imputation models on an independent test dataset obtained from Rosati *et al.*²². (a) shows the balanced accuracy, while (b) the recall and (c) the precision across different HLA alleles belonging to different HLA loci. Across all panels, alleles with carriership frequency <0.05 (n<12 samples) were excluded from the analysis. **The data supporting panels (a-c) are provided in table S4.**

Figure 3 : The performance of the TRB-based HLA imputation models on an independent test dataset obtained from the immuneCODE dataset²³. **(a)** shows the balanced accuracy, while **(b)** the recall and **(c)** the precision across different HLA alleles belonging the different HLA loci. Across all panels, alleles with carrier frequency <0.05 (n<3 samples) were excluded from the analysis. **The data supporting panels (a-c) are provided in table S5**

Figure 4: The performance of the developed TRA-based imputation models on a test dataset of paired TRA repertoire and HLA alleles that was generated by Rosati *et al.*²². **(a)** shows the balanced accuracy, while **(b)** the recall and **(c)** the precision across different HLA alleles belonging to different HLA loci. Across all panels, alleles with carriership frequency <0.05 (n<12 samples) were excluded from the analysis. **The data supporting panels (a-c) are provided in table S6**

2.17 Add a table/figure showing the overall performance of each model w.r.t. each dataset (incl the 20% test set), as it is hard to read the overall performance from figs 2-5. (see also Issue 1.2)

We have annotated the figures with the allele naming showing the performance of the models on each dataset. We also provide supplementary tables (Table S1, S4, S5 and S6) showing the performance of the models on each of the described validation and test datasets.

2.18 Line 332: "it still represents only of the TRB-HLA dataset": rephrase. We have edited the sentence to "it still represents only (<25%) of the TRB-HLA dataset"

2.19 Highlight the novelty of TRA-HLA models in the introduction? Even if it does not perform so well? Perhaps that is an interesting result by itself?

We here wanted to discuss the TRA-HLA models because currently TRB-based model are more accurate and TRA-HLA models are still in their infancy. Hence, discussing the advantages and limitation of this approach is a good fit for the discussion rather than highlighting the advantages in the introduction.

2.20 Does the full model use some kind of regularization? Other authors have included this. Is the model over-fitting? How do the predictions for observations in the training dataset compare to predictions in the test dataset?

The performance on independent test datasets is definitely the most important metric to evaluate the model performance and as discussed in the manuscript, we test the performance of the generated models, particularly the TRB-based models, on two independent test datasets to obtain an estimation of the model performance on real-world data. Regarding the question about regularization: we here used a different strategies to control the model performance at different stages of the pipeline. Some of the statistical models used regulations such as L1-based regularization, however, our final logistic regression that estimates the carriership of an allele uses only two input features. First, the total number of clonotypes present in the repertoire, which is the repertoire depth, and second, the cumulative expansion of allele associated clonotypes in a given repertoire. Hence, for such a simple model and our large dataset size, regularization might decrease the model performance, hence, we did not use regularization here which aggress was what described before for example by the Zahid *et al.* study.

2.21 Line 603: "disease associated clonotypes" should this be "HLA-associated"?

We have edited the manuscript accordingly.

2.22 Can you comment on the choice of the FET? One problematic assumption of this test is that the marginals of the contingency table are fixed. Is this a valid assumption in your case?

Thank you for this thoughtful comment. We would like to clarify our methodological choices. We used Fisher's Exact Test (FET) to assess the association between a given HLA allele carrier status and clonotype carriership status using 2x2 contingency tables. While FET assumes fixed marginals, a condition not strictly satisfied in our observational setting, we selected it due to its

robustness and exact p-value calculation, particularly becoming important given the sparsity and imbalances in some tables.

Crucially, the associations identified via FET were not interpreted in isolation. They served as input features for machine learning models, whose performance was validated on independent test datasets. This empirical evaluation supports the practical utility of the associations, regardless of the marginal assumption underlying FET. We did not use the chi-squared test, as it is less reliable in cases of low or uneven cell counts without appropriate corrections. FET is more suitable under these conditions. We also considered logistic regression; however, our goal was not to estimate effect sizes or control for covariates, but to perform large-scale binary association testing in a statistically robust and interpretable manner. FET offered a practical balance of rigor and computational efficiency for this task.

2.23 Line 615-632: Please give more details. Your method appears to be a bit different from the one in Zahid *et al*, so it would help to have e.g. equations that define the L1-model and the logistic classifier. Also, the methods in Zahid *et al* are not very clear (and I don't think their Eq (1) is correct. See <https://www.biorxiv.org/content/10.1101/2024.04.01.587617v2.full>).

We have edited, extended and revised our methods section as follow: “To resolve this, we followed the L1 regularized linear regression framework proposed by Zahid *et al*.¹³ which was implemented in TensorFlow to increase computational performance and summarized in the following set of equations.

For each clonotype c we have the function L which returns the number of alleles associated with clonotype c , hence, $L(c)$ return an integer that represent the number of HLA alleles that were associated with c . Following this we can construct the indicator function S , defined as follow:

$$S(c) = \begin{cases} 0, & \text{if } L(c) = 1 \\ 1, & \text{if } L(c) > 1 \end{cases}$$

For cases where $S(c) = 1$, we conducted an L1LR analysis by constructing the following matrices for each clonotypes:

1. The label Y vector which is $\in \mathbb{N} \times 1$ where N represent the number of samples and elements in Y are defined as follow

$$Y_i = \begin{cases} 0, & \text{if } D(c, R_i) = 0 \\ 1, & \text{if } D(c, R_i) = 1 \end{cases}$$

Where the $D(c, R_i) = 1$ if clonotype c is detected in the repertoire of the i^{th} sample, otherwise the function returns 0.

2. The input features matrix $X \in R^{N \times (L(c)+1)}$ where N represent the number of samples and the first $L(c)$ columns are indicator columns that indicate if the i^{th} samples have the j^{th} element of the alleles associated with c . The last dimension of the matrix X represents the normalized number of clonotypes defined as follow:

$$X_{i,d+1} = \sqrt{C(R_i)}$$

Where $C(R_i)$ is a function that returns the total number of clonotypes in the i^{th} repertoire.

After constructing these X and Y matrices, we used a logistic regression function to predict the presence of clonotype c in the i^{th} repertoire from the set of associated alleles and normalized number of clonotypes defined in the feature matrix X using a logistic regression function defined as follow:

$$Y_i^{pred} = \frac{1}{1 + e^{-(XW+b)}}$$

Where W represent the weight matrix and b represents the biases of the models, the values of these learned parameters were learned using gradient decent with the following loss function

$$Loss(W, b) = -\frac{1}{n} \sum_{i=1}^n [y_i \log(Y_i^{pred}) + (1 - y_i) \log(1 - Y_i^{pred})] + \lambda \sum_{j=1}^{d-1} |w_j|$$

Where λ is a non-learnable parameter, that is, it is not optimized or learned during the gradient descent. This L1 loss term, $\sum_{j=1}^{d-1} |w_j|$, is used to force enforce sparsity. We keep training the model until all the $w_j < 1 \times 10^{-4}$ except one, where the remaining $w_j > 1 \times 10^{-4}$ being the weight of the allele this clonotypes is mostly associated with. For a given λ , we try running a gradient-decent based optimizer for 1000 epochs, if we could not resolve the HLA association after these 100 epochs, the training loop is restarted with a new value for λ . Ten values for λ were tried ranging from 0.01 to 10.

Subsequently, we arrange the dataset into HLA-alleles and their corresponding HLA associated clonotypes. For each clonotype in the set we can calculate a weight that can be interpreted as the strength of its association with its cognate allele following the same recipe described by Zahid *et al.*¹³. Specifically, we built two matrices for each clonotype c which is associated to an allele

a

1. the input matrix $X \in R^{N \times 2}$ where N represents the number of samples and $X_i = [C(c, R_i), \sqrt{C(R_i)}]$ where $C(R_i)$ represents the total number of clonotypes in the i^{th} sample and the $C(c, R_i)$ represents the expansion of clonotype c in the i^{th} repertoire. Subsequently, we normalized the matrix X using Z-score normalization applied column-wise.

2. the label matrix $Y \in R^{N \times 1}$ which is defined according to the following indicator function

$$Y_i = f(x) = \begin{cases} 1, & \text{if } S_i \text{ carries allele } a \\ 0, & \text{otherwise} \end{cases}$$

After that we train a standard logistic regression model to predict the carriership of the allele a from the expansion of clonotype c in a repertoire R profiled with a given depth. The β_1 weight of this logistic regression model is the weight of clonotype c with regard to predicting allele a .

These weights were used to calculate a weighted sum of the expansion of allele-associated clonotypes in each repertoire. Lastly, a logistic regression classifier to classify repertoires into allele carrier and non-carriers was developed. Specifically, we build two matrices to train logistic regression models to impute carriership alleles from the TRA or the TRB repertoire:

1. the input matrix $X \in R^{N \times 2}$ where N represents the number of samples and $X_i = [\sum_{j=1}^{|A|} |a_j| * C(c_j, R_i), C(R_i)]$ where the $\sum_{j=1}^{|A|} |a_j| * C(c_j, R_i)$ represents a weighted sum of the weighted allele associated clonotypes. A represents a set made of tuples, $\{(a_j, c_j), \dots\}$ where a_j represents the weight of the j^{th} clonotype that is associated with a given allele and c_j is the j^{th} clonotype that is associated with a given allele. The term, $C(c_j, R_i)$, represents the expansion or the count of the j^{th} clonotype in the i^{th} repertoire. Lastly, $|A|$ represents the number of elements in the set of clonotypes associated with a given allele.

2. the label matrix $Y \in R^{N \times 1}$ which is defined according to the following indicator function

$$Y_i = f(x) = \begin{cases} 1, & \text{if } S_i \text{ carries allele } a \\ 0, & \text{otherwise} \end{cases}$$

After that we train a standard logistic regression model to predict the carriership of the allele a from the expansion of clonotype c in a repertoire R profiled with a given depth."

2.24 Fig S1: Typo: "Pearson correlation co-efficiency"

We thank the Reviewer for catching this point, figure S1 has been merged with Figure 2 in agreement with reviewer 1 recommendation and we have corrected the typo in the legend.

2.25 Fig S3: what does the colormap indicate? Please remove. (also in other SI figures)

We have edited and revised all the supplementary figures accordingly in the revised manuscript.

A point-by-point response letter to Reviewer 2 comments

Dear Editor and Reviewer,

We sincerely thank Reviewer 2 for the constructive feedback and for recognizing the improvements made in the revised manuscript. We also appreciate the Editor's efforts in overseeing this process. In response to the reviewer's concerns, we have carefully revised the manuscript for grammar, spelling, and overall structure, with particular attention to the Methods section, which we agree required substantial refinement.

Below, we provide a detailed, point-by-point response to each of the reviewer's comments.

1. I don't completely agree with how the Ortega et al issue was handled: The authors now write: "However, all mentioned studies with the exception of Ortega et al., did not release a publicly available tool for imputing HLA alleles from TCR repertoires". My question is simply: why should we use your tool instead of Ortega et al? (the error you got while running their method can be easily resolved. See my comments below). Running HLAGuessr: The error is not raised if you add the option "-d tab" in the command line. I've added a comment to the issue on GitHub, with a suggestion how to solve the bug.

Thanks for your help in resolving the HLAGuessr bug. We ran the tool and benchmarked it against our TCR2HLA models, finding comparable performance across common HLA alleles. In some cases, TCR2HLA performed better, while in others HLAGuessr had the advantage, depending on the metric used, for example, TCR2HLA often showed higher precision, whereas HLAGuessr achieved better recall. Overall, both tools performed similarly. However, TCR2HLA retains two key advantages: broader allele support and the ability to impute functional HLA-DQ and HLA-DP alleles. We have incorporated these results into the manuscript, adding the following section:

“

TCR2HLA achieves a state-of-the-art performance in imputing HLA allotypes from TRB repertoires

To evaluate the performance of the developed tools against other TCR-based imputation pipelines, we benchmarked their predictive performance against HLAGuessr which was recently developed by Ortega *et al.*¹² focusing on analyzing the performance of the TRB models. We selected the immuneCODE datasets²⁵ as it was not included in the training of our tool (TCR2HLA; Data availability) or that of Ortega and colleagues¹². Beside the difference in predictive performance (**Fig. 6**), TCR2HLA offers two advantages over HLAGuessr, first, it can impute functional HLA-DQ and HLA-DP alleles and not just a chain level prediction, *i.e.* a specific DQA or DQB allele as HLAGuessr does. Second, it supports a larger number of alleles, specifically, 433 alleles, including >175 common HLA alleles, relative to 98 alleles supported by HLAGuessr. As HLAGuessr does not predict functional HLA-DQ and HLA-DP alleles we restricted the comparisons to the three HLA-I loci and the HLA-DRB1 locus focusing on common alleles. Across all loci we observed a comparable performance, where for some alleles the performance of

TCR2HLA was better while for others HLAGuessr yielded a better performance, e.g. HLA-A*11:01 and HLA-A*03:01, respectively (**Fig. 6**). Furthermore, the tools also differ across metrics, for example, the TCR2HLA model for the HLA-B*14:01 allele showed a higher precision while the HLAGuessr's model of the same allele showed higher recall and balanced accuracy (**Fig. 6**). Thus, for common alleles, both tools achieved comparable performance, consequently, future tools can be developed to combine predictions by both tools to improve the overall imputation accuracy of HLA alleles from TCR datasets.

Figure 1: benchmarking the predictive performance of TCR2HLA against HLAGuessr using the immuneCODE datasets. (a-c) the performance of HLA models across three metrics, namely, balanced accuracy, precision and recall, respectively. (d-f) the performance across the three evaluation-metrics for HLA-B proteins, (g-i) and (j-l) the benchmarking results for HLA-C and HLA-DRB1 proteins respectively. The supporting data is available in table S7.”

Here are some specific points:

1. Intro: Thank you for adding this final paragraph and explaining the importance of your work. However, this paragraph could be better integrated with the rest of the introduction.

We have restructured the introduction to better highlight the importance of the work, as shown below and in the revised manuscript.

“

These studies have established the feasibility of performing HLA typing from the T cell repertoire as well as the utility of large-scale statistical analyses in identifying public clonotypes associated with different HLA alleles. However, all mentioned studies **apart from Ortega *et al.*¹²**, did not release a publicly available tool for imputing HLA alleles from TCR repertoires. **As T cell repertoire analyses became an import method to understand tumors^{14,15}, response to infectious diseases^{16,17} and identifying antigens driving chronic inflammatory diseases^{18–20}, the developed imputation models can decrease the cost of these analyses by removing the need to perform laborious and expensive wet-lab-based HLA-typing. Furthermore, these models can impute only functional HLA proteins shaping the repertoire which is especially important for HLA-DQ and HLA-DP proteins because both the α and the β chains are polymorphic however, not all assembled $\alpha\beta$ heterodimers produce a functional protein. Lastly, the generated datasets of TRA and TRB clonotypes associated with their cognate HLA allele, is of paramount importance for analyzing disease-associated clonotypes, which are a group of public clonotypes that are expanded in a specific disease¹⁸, for example, primary sclerosis cholangitis (PSC)¹⁹ or inflammatory bowel disease (IBD)^{20–22}. To produce publicly available tools to impute HLA allotypes from either the TRA or the TRB repertoire and generate a database of public TCRs with their cognate HLA allotype, we assembled a large dataset of 1,240 TRA and 5,554 TRB repertoires with their HLA allotypes, enabling us to identify the cognate HLA alleles for 34,206 TRA and 891,564 TRB clonotypes from 175 common HLA alleles. After that we utilized the identified TRA and TRB clonotypes to develop machine-learning models that can be used to impute the carriership of these HLA alleles.”**

2. Page 6: "however, for a small subset of clonotypes this was not possible". Thank you for explaining this to me. However, I do think that adding a very brief explanation in the text will be helpful. You could mention when the algorithm fails in the Online methods section and in the text say: "however, for a small subset of clonotypes this was not possible (see Online Methods)"

We have edited the method section accordingly as depicted below and in the manuscript:
“

After constructing these X and Y matrices, we used a logistic regression function to predict the presence of clonotype c in the i^{th} repertoire from the feature matrix X as follow:

$$Y_i^{\text{pred}} = \frac{1}{1 + e^{-(XW+b)}}$$

Where W represents the learned weights and b the biases of the model, these parameters were learned using gradient descent with the following loss function

$$\text{Loss}(W, b) = -\frac{1}{n} \sum_{i=1}^n [y_i \log(Y_i^{\text{pred}}) + (1 - y_i) \log(1 - Y_i^{\text{pred}})] + \lambda \sum_{j=1}^{d-1} |w_j|$$

Where λ is a non-learnable parameter, *i.e.* it is not learned using gradient descent but has a fixed value. This L1 loss term, $\sum_{j=1}^{d-1} |w_j|$, is used to enforce sparsity. We keep training the model until all the $w_j < 1 \times 10^{-4}$ except one, this remaining $w_j > 1 \times 10^{-4}$ being the weight of the allele this clonotypes is mostly associated with. For a given λ , we try running a gradient-descent based optimization for a 1000 epochs, if we could not resolve the HLA association after these 100 epochs, that is, there is more than one weight $> 1 \times 10^{-4}$, then the training loop is restarted with a new value for λ . Ten values for λ were tried ranging from 0.01 to 10. If we could not achieve the required optimization task of getting all the weights to be $< 1 \times 10^{-4}$ except one with these ten different λ values, then we label the input clonotype as “unsolvable”. This imply that this clonotype is strongly linked with a specific HLA haplotype and with the current dataset in term of sample size and population structure we cannot resolve the association to a specific allele within the associated haplotype. These clonotypes are filtered from all downstream analyses and steps.”

3. Figure 3: maybe you can remove the locus from the HLA type in the annotations? (it is a bit crowded) so write 03:03 instead of DQA1*03:03

We have edited the figures accordingly including Figure 3, 4 and 5 to address this point as shown below and in the revised manuscript. ”

Figure 2: The performance of the TRB-based HLA imputation models on an independent test dataset obtained from Rosati *et al.*²⁴. (a) shows the balanced accuracy, while (b) the recall and (c) the precision across different HLA alleles belonging to different HLA loci. Across all panels, alleles with carriership frequency <0.05 ($n < 12$ samples) were excluded from the analysis. The data supporting panels (a-c) are provided in table S4.

Figure 3 : The performance of the TRB-based HLA imputation models on an independent test dataset obtained from the immuneCODE dataset²⁵. (a) shows the balanced accuracy, while (b) the recall and (c) the precision across different HLA alleles belonging the different HLA loci. Across all panels, alleles with carriership frequency < 0.05 ($n < 3$ samples) were excluded from the analysis. The data supporting panels (a-c) are provided in table S5.

Figure 4: The performance of the developed TRA-based imputation models on a test dataset of paired TRA repertoire and HLA alleles that was generated by Rosati et al.²⁴. (a) shows the balanced accuracy, while (b) the recall and (c) the precision across different HLA alleles belonging to different HLA loci. Across all panels, alleles with carriership frequency <0.05 ($n < 12$ samples) were excluded from the analysis. The data supporting panels (a-c) are provided in table S6.”

- page 16: "Although the assembled TRA-HLA dataset reported here represents, to the best of our knowledge, the biggest dataset of its kind, it still represents only (<25%) of the TRB-HLA dataset". I'm sorry, but this sentence still does not make sense. Maybe write: "Although the assembled TRA-HLA dataset reported here is (to the best of our knowledge) the largest dataset of its kind, it is still relatively small (<25%) compared to the TRB-HLA dataset."

Have been edited correctly.

- Online Methods; Developing the TCR-based HLA imputation framework: Thank you for explaining the method in detail. I think it would help to check this text for redundancy and grammar, though. For instance, the phrase is "as follows" (not "as follow"). Also, instead of writing "For each clonotype c we have the function L which

returns the number of alleles associated with clonotype c , hence, $L(c)$ return an integer that represent the number of HLA alleles that were associated with c ." you can simply say "For each clonotype c , we let $L(c)$ denote the number of HLA alleles associated with c ." and instead of "Following this we can construct the indicator function S , defined as follow: $S(c) = \dots$ For cases where $S(c) = 1$, we conducted an L1LR analysis..." you can just write: "For clonotypes c with $L(c) > 1$ we conducted an L1LR analysis..." There's no need to introduce an indicator function S . There are a lot of needlessly convoluted sentences in this section, so please read through this part again and simplify.

6. page 23; point 2: What is S_i ? Subject i perhaps? Please clarify (or just say "if subject i carries allele a). Also: what is $f(x)$? Maybe remove this? Again, check spelling/grammar: "logistic region model".
7. Especially the last paragraph of page 23 is rather incomprehensible to me. Please fix this. For example, why do you take the absolute value of the weight $|a|$

Thanks for the recommendations and the feedback. For **point 5, 6 and 7** we have edited and revised the methods section accordingly as depicted below and in the revised manuscript: “

For each clonotype c , we let $L(c)$ denote the number of HLA alleles associated with c , for clonotypes with $L(c) > 1$ we conducted an L1LR analysis by constructing two matrices for each clonotype:

1. The label Y matrix which is $\in N \times 1$ where N represents the number of samples and elements in Y are defined as follow

$$Y_i = \begin{cases} 1, & \text{if clonotype } c \text{ is detected in the repertoire of the } i^{\text{th}} \text{ sample} \\ 0, & \text{otherwise} \end{cases}$$

2. The input features matrix $X \in R^{N \times (L(c)+1)}$ where N represents the number of samples and the first $L(c)$ columns are indicator columns, indicating if the i^{th} individual carries the j^{th} HLA allele associated with c . The last dimension of the matrix X represents the normalized number of clonotypes defined as follow:

$$X_{i,d+1} = \sqrt{C(R_i)}$$

Where $C(R_i)$ is the total number of clonotypes in the i^{th} repertoire.

After constructing these X and Y matrices, we used a logistic regression function to predict the presence of clonotype c in the i^{th} repertoire from the feature matrix X as follow:

$$Y_i^{pred} = \frac{1}{1 + e^{-(XW+b)}}$$

Where W represents the learned weights and b the biases of the model, these parameters were learned using gradient descent with the following loss function

$$Loss(W, b) = -\frac{1}{n} \sum_{i=1}^n [y_i \log(Y_i^{pred}) + (1 - y_i) \log(1 - Y_i^{pred})] + \lambda \sum_{j=1}^{d-1} |w_j|$$

Where λ is a non-learnable parameter, *i.e.* it is not learned using gradient descent but has a fixed value. This L1 loss term, $\sum_{j=1}^{d-1} |w_j|$, is used to enforce sparsity. We keep training the model until all the $w_j < 1 \times 10^{-4}$ except one, this remaining $w_j > 1 \times 10^{-4}$ being the weight of the allele this clonotypes is mostly associated with. For a given λ , we try running a gradient descent based optimization for a 1000 epochs, if we could not resolve the HLA association after these 100 epochs, that is, there is more than one weight $> 1 \times 10^{-4}$, then the training loop is restarted with a new value for λ . Ten values for λ were tried ranging from 0.01 to 10. If we could not achieve the required optimization task of getting all the weights to be $< 1 \times 10^{-4}$ except one with these ten different λ values, then we label the input clonotype as “unsolvable”. This imply that this clonotype is strongly linked with a specific HLA haplotype and with the current dataset in term of sample size and population structure we cannot resolve the association to a specific allele within the associated haplotype. These clonotypes are filtered from all downstream analyses and steps.

Subsequently, we arrange the dataset into HLA alleles with their associated clonotypes. For each clonotype in this set we calculated a weight that indicates the strength of its association with its cognate HLA allele following the same approach described by Zahid *et al.*¹³. Specifically, we built two matrices for each clonotype c which is associated to an allele a

1. the input matrix $X \in R^{N \times 2}$ where N represents the number of samples and $X_i = [C(c, R_i), \sqrt{C(R_i)}]$ where $C(R_i)$ returns the total number of clonotypes in the i^{th} samples and the $C(c, R_i)$ represents the expansion of clonotype c in the i^{th} repertoire. Subsequently, we normalized the matrix X using Z-score normalization applied column-wise.

2. the label matrix $Y \in R^{N \times 1}$ which is defined according to the following indicator function

$$Y_i = \begin{cases} 1, & \text{if Subject}_i \text{ carries allele } a \\ 0, & \text{otherwise} \end{cases}$$

After that we train a logistic regression model to predict the carriership of the allele a from the expansion of clonotype c in a repertoire R profiled at a given depth. The absolute value of the β_1 weight of this model is the weight of clonotype c that is associated with the HLA allele a .

These weights were used to calculate a weighted sum of allele-associated clonotypes' expansion in each repertoire. Lastly, a logistic regression classifier was developed to predict carrier and non-carrier of a given allele based on this weighted expansion. Specifically, we build two matrices to train these models:

1. the input matrix $X \in R^{N \times 2}$ where N represents the number of samples and $X_i = [\sum_{j=1}^{|A|} |a_j| * C(c_j, R_i), C(R_i)]$ where the $\sum_{j=1}^{|A|} |a_j| * C(c_j, R_i)$ represents a weighted sum of the expansion of allele-associated clonotypes A . This set is made of tuples, $\{(a_j, c_j), \dots\}$ where a_j represents the weight of the j^{th} clonotype that is associated with a given allele and c_j is the j^{th} clonotype that is associated with the same allele. The term, $C(c_j, R_i)$, represents the expansion of the j^{th} clonotype in the i^{th} repertoire. The $|a_j|$ term represents the absolute value of the clonotype weight identified by logistic regression as shown above, because we focused on the strength of association independent of its direction. Hence, if an allele a is associated with three clonotypes, c_1, c_2 and c_3 its set of associated clonotypes will be arranged as follow: $\{(w_1, c_1), (w_2, c_2), (w_3, c_3)\}$, where w is the weight and c is the clonotype. To calculate the weighted expansion of the allele associated clonotypes in the i^{th} repertoire, we can use the expanded form of the equation described above as follow: $|w_1| * \text{expansion of } c_1 \text{ in } R_i + |w_2| * \text{expansion of } c_2 \text{ in } R_i + |w_3| * \text{expansion of } c_3 \text{ in } R_i$. Lastly, $C(R_i)$ is the total number of unique clonotypes present in the i^{th} repertoire.

2. the label matrix $Y \in R^{N \times 1}$ defined according to the following indicator function

$$Y_i = \begin{cases} 1, & \text{if Subject}_i \text{ carries allele } a \\ 0, & \text{otherwise} \end{cases}$$

After that we train a standard logistic regression model to predict the carriership of the allele a in a repertoire R profiled at a given depth using the allele-associated clonotypes set A defined above.”

8. Figure 1C: "cumulative expansion of allele-associated clonotype set"
Thanks for explaining this. However, my suggestion was to explain this in the actual figure

caption. In the caption, you are referring to a "weighted sum" instead (which is more accurate IMO).

We have edited the caption as follow: "(c) a summary of building a classifier to predict the carriership of a given HLA allele using the cumulative expansion of allele-associated clonotype set for that allele in a given TRA or TRB repertoire."

- 9. Point 1.2: I've tried your model on the provided data, and often you get e.g. 2 HLA-A alleles with high probability. This makes sense for heterozygous individuals. Therefore, I don't understand your response 2: "To provide a normalization form for the calculated P-value...". You're using softmax on the probabilities from a single locus. If someone has alleles A*01:01 and A*01:02, and the model correctly calls those with high probability (e.g. 0.98 and 0.99, and 0.00001 for the others), then the softmax would result in approx $p = 0.5$ for A*01:01 and $p = 0.5$ for A*02:01. That does not make sense to me. Are you using this normalization for the results in the paper? If so, I don't think you're explaining this.*

We would like to clarify that the SoftMax normalization was not used in generating any of the results presented in the manuscript. It was introduced only in response to your earlier comment regarding probability normalization across different prediction modules. By design, the normalization step weights the probability of each HLA prediction in a given locus, but the default behavior of the tool is to predict carriership for each allele independently and to report the probability of carrying that allele across all supported HLA alleles.